# DICEFORMER: SPIKING AUDIO TRANSFORMER WITH DENSITY-AWARE DICE ATTENTION

## ABSTRACT

Spiking Neural Networks (SNNs) have garnered significant attention due to their potential for low energy consumption. However, their application in the audio domain remains relatively underexplored. This work aims to close this gap by designing spiking transformers suitable for audio processing applications. We introduce DiceFormer, a directly trained spiking transformer that incorporates two novel components: (i) Spike Dice Attention (SDA), a spike-based attention module that leverages the Dice similarity concept to produce density-aware attention scores, which improve the modeling of spike-based representations; and (ii) Spike Audio Dice Attention (SADA), an SDA-based extension specifically designed to handle the frequency–temporal features inherent in complex audio spectrograms. Extensive experiments demonstrate that DiceFormer achieves superior performance over existing state-of-the-art (SOTA) SNNs on mainstream audio datasets. Notably, when trained from scratch, DiceFormer achieves an mAP of 0.161 on AudioSet (20K) with only 54.3M parameters, substantially outperforming prior models. It also establishes new SOTA results on ESC-50 and SCV2, highlighting the promise of SNNs in complex audio processing.

## 1 INTRODUCTION

Artificial Neural Networks (ANNs) especially the Transformer architecture have propelled major advances across computer vision, audio, and speech. In computer vision, Transformer-based models drive progress in image classification (Yuan et al., 2021; Liu et al., 2021), object detection (Carion et al., 2020; Zhu et al., 2020), and semantic segmentation (Wang et al., 2021; Yuan et al., 2022); in audio, they enable sound event classification (Gong et al., 2021; 2022; Huang et al., 2022), source separation (Xu et al., 2025; Subakan et al., 2021); and in speech, large attention-based encoders have become standard for automatic speech recognition and representation learning (Baevski et al., 2020; Hsu et al., 2021; Radford et al., 2023). However, as the scale and complexity of ANN models increase, they face a significant challenge of substantial energy consumption. To address this, Spiking Neural Networks (SNNs) (Maass, 1997; Izhikevich, 2003; Masquelier et al., 2008) (Appendix A for details) have emerged as a promising alternative, offering an event-driven computational paradigm inspired by the human brain. Operating with discrete spikes, SNNs enable much greater energy efficiency, motivating their use in applications ranging from bearing fault diagnosis (Lim & Kim, 2025b) to biosignal processing (Kang et al., 2025). However, despite this promise, SNNs have consistently exhibited lower accuracy compared to ANNs, and recent progress in bridging this gap has been predominantly concentrated in the image domain.

Much of this progress stems from adapting Vision Transformer (ViT) (Dosovitskiy et al., 2021) architectures to the spiking setting. In image classification, models such as Spikformer (Zhou et al., 2023), Spike-driven Transformer (Yao et al., 2023), and QKFormer (Zhou et al., 2024) have achieved strong performance while retaining energy efficiency. Similarly, Spike-driven Transformer V2 (Yao et al., 2024) extends these gains to dense prediction tasks such as semantic segmentation and object detection. Further successes have been reported in generative modeling, with spiking variational autoencoders (VAEs) (Kamata et al., 2022) and spiking diffusion models (Cao et al., 2024).

Despite these advances, progress has remained largely confined to the visual domain, and the optimization of attention mechanisms for spike signals is still not well established. Existing spike self-attention mechanisms, including dot-product (Zhou et al., 2023; Yao et al., 2024) and Hadamard

product (Yao et al., 2023), are density-unaware, as attention scores are computed without considering the spike density of query and key vectors. This can cause attention scores to be biased toward vectors with high spike density, as the mechanism cannot distinguish between dense spiking activity and true signal similarity. Consequently, the model may misinterpret raw spike counts as indicators of similarity, limiting its ability to capture genuine spike-based relationships. Moreover, dot-product-based formulations incur quadratic complexity, leading to a substantial increase in computation as the model scales up. Beyond the limitation of attention mechanisms, applying spike-based transformers to audio requires architectures explicitly tailored to audio spectrograms. Current SNN-ViT models are primarily designed for images and are therefore not well-suited to capture the frequency-temporal correlations of audio data. To the best of our knowledge, few prior works have introduced SNN architectures explicitly designed for audio classification and validated across diverse audio datasets.

To overcome these limitations, we propose **DiceFormer**, a novel SNN architecture optimized for audio processing tasks. At the core of DiceFormer are two components: **Spike Dice Attention (SDA)**, a linear-time attention mechanism that is robust to spike-density variation, and the **Spike Audio Dice Attention (SADA)**, an extension of SDA that decouples frequency and temporal axes to effectively learn audio-specific frequency-temporal features. We evaluate DiceFormer on three widely used audio classification benchmarks: AudioSet (Gemmeke et al., 2017), ESC-50 (Piczak, 2015), and Speech Commands V2 (Warden, 2018). Trained from scratch, DiceFormer achieves a new state-of-the-art (SOTA) performance among SNNs of 0.161 mAP on AudioSet (20K) and establishes new SNN benchmark performances on the other audio datasets. Furthermore, compared to baseline ANN models across these benchmarks, DiceFormer reduces parameters and energy consumption while maintaining competitive performance. This achievement highlights the effectiveness of our architectural design and establishes a new benchmark that will facilitate future advancements in SNN research for audio processing, providing a pathway toward broader adoption of spiking models in real-world applications.

Our main contributions are summarized as follows:

1. **Spike Dice Attention (SDA).** We introduce SDA, a novel attention mechanism for SNNs that explicitly accounts for spike density. SDA produces robust similarity estimates across varying spike activities while maintaining $O(ND)$ linear complexity. We also provide both theoretical analysis and empirical validation of its effectiveness.

2. **Spike Audio Dice Attention (SADA).** We develop SADA, extended from SDA, tailored for audio processing by decoupling frequency and temporal processing. SADA enables the model to effectively capture the frequency-temporal features.

3. **Bridging the Performance Gap to ANNs.** DiceFormer achieves new SOTA performance among SNNs on key audio benchmark datasets. Trained entirely from scratch, it narrows the long-standing performance gap between SNNs and ANNs while delivering superior energy efficiency.

## 2 RELATED WORK

**Audio Transformers.** Early progress in audio classification was driven by CNN-based architectures such as PANNs (Kong et al., 2020), which provided strong baselines through convolutional feature extraction. However, the field has undergone a paradigm shift toward Transformer-based architectures, which are better suited to capture long-range dependencies in sequential data. A pioneering step in this direction was the Audio Spectrogram Transformer (AST) (Gong et al., 2021), which adapted the Vision Transformer (ViT) paradigm to audio by treating spectrograms as two-dimensional images. AST divides the spectrogram into overlapping patches and processes them sequentially, enabling the model to capture global frequency–temporal correlations. Additionally, SSAST Gong et al. (2022) is an Audio Spectrogram Transformer that performs self-supervised pre-training using Masked Spectrogram Patch Modeling (MSPM), which involves masking spectrogram patches and then reconstructing or contrasting them. Building on this idea, subsequent works have introduced designs that better align with the unique properties of audio data. For example, HTS-AT (Chen et al., 2022) employed a hierarchical structure that integrates information across multiple scales, thereby improving the representation of both local and global features. More recently, DTF-AT (Alex et al., 2024) explicitly decoupled temporal and frequency modeling, creating a dual-branch

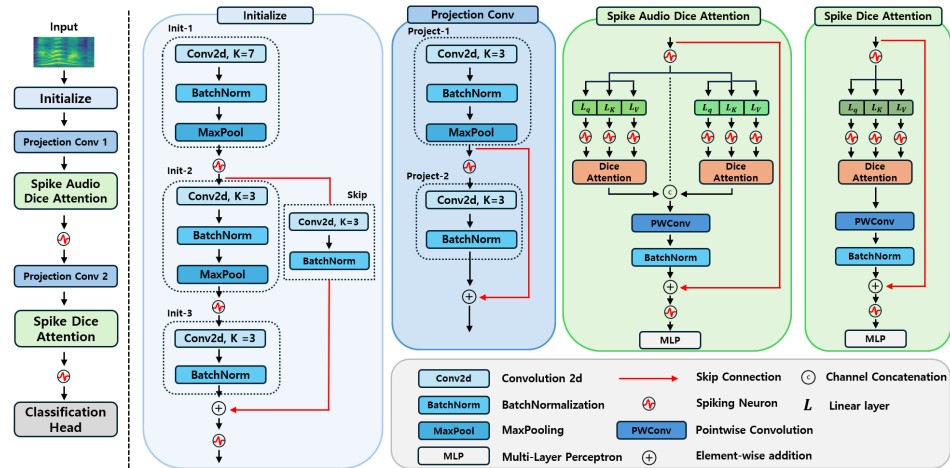

Figure 1: The overall architecture of our proposed DiceFormer. The model processes the input through five main stages: an Initialize block, Projection Conv blocks, a SADA module, a SDA module, and a final Classification Head.

architecture that further advanced performance by learning these two dimensions separately before combining them into a unified representation.

**SNN Vision Transformers.** The adaptation of the Vision Transformer (ViT) (Dosovitskiy et al., 2021) to Spiking Neural Networks (SNNs) has spurred a new generation of high-performance spiking architectures. Spikformer (Zhou et al., 2023) was the first to propose an SNN-ViT, introducing dot-product-based spiking self-attention that demonstrated the feasibility of combining spiking dynamics with Transformer architectures. Building on this, Spike-driven Transformer (Yao et al., 2023) proposed a Hadamard-product-based attention mechanism, achieving linear computational complexity and significantly narrowing the performance gap between SNNs and ANNs on image classification. The framework was later extended by Spike-driven Transformer V2 (Yao et al., 2024), which introduced a hierarchical architecture combined with dot-product attention to achieve strong results in dense prediction tasks, including object detection and semantic segmentation. Most recently, QKFormer (Zhou et al., 2024) advanced this line of research by combining hierarchical representations with a more efficient attention mechanism, achieving state-of-the-art accuracy for SNNs on ImageNet-1K (Deng et al., 2009) and demonstrating performance competitive with widely used ANN baselines such as ViT (Dosovitskiy et al., 2021) and DeiT (Touvron et al., 2021).

## 3 METHOD

The architecture of our proposed DiceFormer, illustrated in Figure 1, is a hierarchical network designed to learn features at multiple scales. As a Spiking Neural Network, DiceFormer integrates Spiking Neurons (SN) after its main computational layers to convert continuous-valued features into binary spikes for event-driven processing.

**Spiking Neuron** To convert continuous-valued features into binary spikes, we employ the Parametric Leaky Integrate-and-Fire (PLIF) neuron model (Fang et al., 2021). Its dynamics at each time-step $t$ are defined as

$$H[t] = V[t-1] + \frac{1}{\tau}\left(X[t] - (V[t-1] - V_{\text{reset}})\right) \tag{1}$$

$$S[t] = \text{Heav}\big(H[t] - V_{\text{th}}\big) \tag{2}$$

$$V[t] = H[t]\,(1 - S[t]) + S[t]\,V_{\text{reset}}, \tag{3}$$

where $X[t]$ is the input current, $V_{\text{th}}$ is a the firing threshold, and $\tau$ is the membrane time constant. $H[t]$ denotes the pre-spike membrane potential obtained by leaky integration of $V[t-1]$ and $X[t]$. The output spike $S[t] \in \{0,1\}$ is generated by the Heaviside step function $\text{Heav}(\cdot)$, which emits 1 when $H[t] \geq V_{\text{th}}$. The post-spike membrane potential $V[t]$ equals $H[t]$ if no spike occurs, and

is reset to $V_{\text{reset}}$ otherwise. The time constant $\tau$ is treated as a learnable parameter, allowing the neuron's temporal dynamics to be optimized during training.

**Overall Architecture** DiceFormer consists of five main components: an Initialize block, Projection Conv blocks, the SADA and an SDA, and a final Classification Head (CH). Given an input spectrogram $I \in \mathbb{R}^{T \times C \times H \times W}$[1], the processing pipeline is as follows. The **Initialize** block maps channels $C \to E$ and downsamples the spatial resolution by a factor of 4 ($H \to H/4$, $W \to W/4$). The first **Projection Conv 1** block further maps channels $E \to D_1$ and applies an additional $\times 2$ downsampling ($H/4 \to H/8$, $W/4 \to W/8$). The **SADA** module then splits the $D_1$ channels into two parallel streams of size $D_1/2$, where frequency attention operates over $N_{\text{freq}} = H/8$ tokens and temporal attention over $N_{\text{temp}} = W/8$ tokens. After this stage, the second **Projection Conv 2** block performs one more downsampling, changing feature maps channels from $D_1 \to D_2$, and the **SDA** module applies unified self-attention on the flattened token sequence $N_{HW} = (H/16) \times (W/16)$ with $D_2$ feature channels. Finally, the SDA output is aggregated by Global Average Pooling (GAP) and passed to the **Classification Head (CH)** for prediction. Implementation details for each module are provided in Appendix B.

### 3.1 INITIALIZE

The Initialize block serves as the CNN stem (Liu et al., 2022) of DiceFormer, performing initial feature extraction and spatial downsampling from the input spectrogram. As shown in Figure 1, it consists of three sequential sub-blocks (**Init-1**, **Init-2**, **Init-3**) and a parallel skip connection (**Skip**). **Init-1** applies a $7 \times 7$ Conv2d, BatchNorm, and MaxPool, followed by a Spiking Neuron (SN). The main path then proceeds through **Init-2**, which applies a $3 \times 3$ Conv2d, BatchNorm, and another MaxPool, again followed by an SN. The final stage, **Init-3**, applies a $3 \times 3$ Conv2d and BatchNorm. In parallel, the **Skip** path—branched from the input of **Init-2**—contains its own $3 \times 3$ Conv2d and BatchNorm to align channel dimensions with those of **Init-3**. The outputs of **Init-3** and **Skip** are then combined by element-wise addition, and the summed feature is passed through an SN. This output, $X_1$, has each spatial dimension reduced by a factor of 4 ($H \times W \to H/4 \times W/4$) and its channel dimension projected to the base embedding size $E$

### 3.2 PROJECTION CONV

The Projection Conv blocks act as transition layers (Huang et al., 2017) in the hierarchical architecture of DiceFormer. Their role is to further downsample feature maps while projecting channels to match the dimensionality required by subsequent attention modules (SADA and SDA). As illustrated in Figure 1, each block consists of two sub-blocks, **Project-1** and **Project-2**. **Project-1** performs downsampling with a $3 \times 3$ Conv2d, BatchNorm, and MaxPool (stride 2), thereby reducing each spatial dimension by half. Its output is then passed through an SN, which provides the input to **Project-2**. The latter applies a $3 \times 3$ Conv2d and BatchNorm without further altering spatial resolution. A skip connection adds the output of **Project-1** (before the SN) to the output of **Project-2**. The combined feature is finally processed by an SN to produce the block output.

### 3.3 DICE-BASED SPIKE ATTENTION

The self-attention mechanism in ANN-based Transformer relies heavily on Softmax normalization of real-valued feature vectors (Vaswani et al., 2017). In SNNs, however, information is represented as event-based binary spikes. Consequently, directly applying Softmax is not only difficult but also fails to exploit the inherent sparsity and event-driven nature of SNNs (Zhou et al., 2023). It is therefore essential in the SNN field to design attention mechanisms specifically tailored to spike signals. Motivated by this need, we analyze the limitations of existing spike-based attention formulations and propose a new attention mechanism tailored for spike processing.

Figure 2 illustrates the two primary approaches for spike-based self-attention. The first is dot-product-based Spike Self-Attention (SSA) with computational complexity $O(N^2 D)$. The second

---

[1]For static 2D inputs ($I_s \in \mathbb{R}^{C \times H \times W}$), we form a length-$T$ sequence by repeating the same frame $T$ times along the temporal axis to match the SNN interface. In practice, the temporal dimension $T$ is treated as an independent axis for spiking neuron layers, while in other layers, it is typically merged with the batch size.

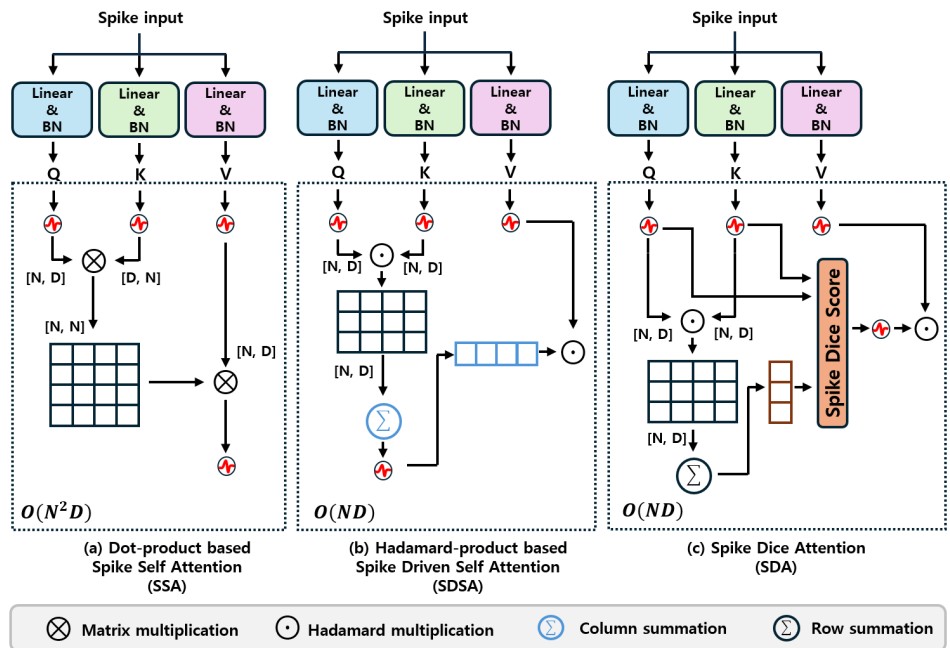

Figure 2: (a) Dot-product–based Spike Self-Attention (SSA, $O(N^2D)$, **density-unaware**); (b) Hadamard-product–based Spike-Driven Self-Attention (SDSA, $O(ND)$, **density-unaware**); (c) Spike Dice Attention (SDA, $O(ND)$, **density-aware**). Here, $N$ is the number of tokens and $D$ the channel dimension.

approach is Spike Driven Self-Attention (SDSA), which replaces dot products with Hadamard Products (HP) followed by summation, reducing the complexity to $O(ND)$. These mechanisms have been widely adopted in recent SNN-ViT models (Zhou et al., 2023; Yao et al., 2024; 2023).

However, both dot-product and Hadamard-product attention suffer from a key limitation: their scores are strongly biased by spike density, yielding an inaccurate measure of true similarity. As illustrated in Figure 3, both density-unaware attention mechanisms assign the same score (3) across four distinct spike keys, treating a sparse, precise match (Key 1) as equivalent to a much denser, less specific one (Key 4). Our empirical analysis further confirms this density bias, showing a strong positive correlation between *conventional spike-attention* scores and overall spike density (see Appendix F).

To overcome this issue, we propose SDA (Appendix C). At its core is the Spike Dice Score

| | Attention Score | | | | | | | | Dot (SSA) | HP (SDSA) | Dice (SDA) |
|---|---|---|---|---|---|---|---|---|---|---|---|
| Spike Query | | | | | | | | | | | |
| Spike Key 1 | | | | | | | | | 3 | 3 | 1 |
| Spike Key 2 | | | | | | | | | 3 | 3 | $\frac{6}{7}$ |
| Spike Key 3 | | | | | | | | | 3 | 3 | $\frac{6}{8}$ |
| Spike Key 4 | | | | | | | | | 3 | 3 | $\frac{6}{9}$ |

Figure 3: An illustrative example of density-aware scoring. SSA (dot) and SDSA (Hadamard + sum) yield the same score despite different key densities, whereas SDA (Dice) differentiates them by reflecting the density difference. *Toy single-token case.*

(SDS), a Dice-based similarity measure that incorporates density-aware normalization (Dice, 1945). This design substantially reduces the correlation between attention scores and spike density (see Appendix F), confirming robustness to density bias. Formally, SDS(Q,K) is defined as follows:

$$\mathrm{SDS}(Q, K) = \frac{2\,\mathrm{sum}_d(Q \odot K)}{\mathrm{sum}_d(Q) + \mathrm{sum}_d(K) + \epsilon} \tag{4}$$

where $\odot$ denotes the element-wise Hadamard product, and $\mathrm{sum}_d$ is the sum over the channel axis and $\epsilon = 10^{-6}$ ensures numerical stability. Because SDS requires only element-wise Hadamard products and channel-wise reductions, it retains linear complexity $O(ND)$.

### 3.3.1 SPIKE AUDIO DICE ATTENTION

The SADA module is designed to learn rich frequency-temporal features by explicitly decoupling and then fusing information from the frequency and temporal axes. The module's operation consists of three main stages.

**1. Input Processing and Decoupling.** Let $X_{\text{in}} \in \mathbb{R}^{T \times D_1 \times H/8 \times W/8}$ be the output of the preceding projection conv block, where $H$ and $W$ denote the frequency and temporal axes, respectively. After passing through a spiking neuron (SN), we obtain $X_{\text{spike}} \in \mathbb{R}^{T \times D_1 \times H/8 \times W/8}$, which is split along the channel dimension into two equal halves ($D_1' := D_1/2$) and routed to a frequency branch and a temporal branch in parallel. The original $X_{\text{in}}$ is retained for the final residual connection applied after feature fusion. In the frequency branch, the temporal axis ($W/8$) is merged into the batch dimension so that each temporal slice is treated as an independent sequence of length $N_{\text{freq}} = H/8$. In the temporal branch, the frequency axis ($H/8$) is merged into the batch dimension so that each frequency slice is treated as an independent sequence of length $N_{\text{temp}} = W/8$.

**2. Parallel Attention Streams.** The two halves of the split tensor, denoted as $X$, are processed in parallel, with one stream operating along the frequency axis and the other along the temporal axis. In both streams, the input is reshaped such that the designated axis serves as the sequence length for attention: $\text{Freq} \in \mathbb{R}^{T \times N_{\text{freq}} \times D_1'}$ with $N_{\text{freq}} = H/8$, and $\text{Temp} \in \mathbb{R}^{T \times N_{\text{temp}} \times D_1'}$ with $N_{\text{temp}} = W/8$.

From each reshaped tensor, query ($Q$), key ($K$), and value ($V$) spike vectors are generated using parallel Linear–BatchNorm–SN branches with learnable matrices:

$$Q = \text{SN}(\text{BN}(XW_Q)), \quad K = \text{SN}(\text{BN}(XW_K)), \quad V = \text{SN}(\text{BN}(XW_V)). \tag{5}$$

The attention output is computed by forming a spike-dice map from the SDS between $Q$ and $K$. This continuous-valued score is passed through an SN to obtain a binary map, which is then applied to $V$ via element-wise (Hadamard) product with channel-wise broadcasting:

$$\text{Attn} = \text{SN}(\text{SDS}(Q, K)) \odot V. \tag{6}$$

Applying this procedure separately yields the frequency-focused output $\text{Attn}_{\text{freq}}$ and the temporal-focused output $\text{Attn}_{\text{temp}}$.

**3. Feature Fusion and Output.** The frequency- and temporal-attention outputs, $\text{Attn}_{\text{freq}}$ and $\text{Attn}_{\text{temp}}$, are concatenated along the channel dimension to reconstruct the original size $D_1$. The combined tensor is processed by a Pointwise Convolution (PWConv) (Howard et al., 2017) with BatchNorm, enabling the model to learn cross-dependencies between frequency and temporal features. A residual connection with $X_{\text{in}}$ is then applied, and the result is passed through a spiking neuron (SN) followed by an MLP block to produce the final output:

$$X_{\text{fused}} = \text{BN}(\text{PWConv}(\text{concat}(\text{Attn}_{\text{freq}}, \text{Attn}_{\text{temp}}))) + X_{\text{in}}, \tag{7}$$
$$X_{\text{out}} = \text{MLP}(\text{SN}(X_{\text{fused}})). \tag{8}$$

### 3.3.2 SPIKE DICE ATTENTION

Whereas the SADA module explicitly decouples frequency and temporal information, the SDA performs unified attention over the combined frequency–temporal dimensions.

Given an input tensor $X_{\text{in}} \in \mathbb{R}^{T \times D_2 \times (H/16) \times (W/16)}$ from the second Projection Conv block, we first flatten the spatial dimensions into a sequence of length $N_{HW} = (H/16) \times (W/16)$, yielding a tensor of shape $T \times N_{HW} \times D_2$. After passing through a spiking neuron (SN), this produces $X_{\text{spike}} \in \mathbb{R}^{T \times N_{HW} \times D_2}$.

From $X_{\text{spike}}$, query ($Q$), key ($K$), and value ($V$) vectors are generated via parallel Linear–BatchNorm–SN branches with learnable matrices:

$$Q = \text{SN}(\text{BN}(X_{\text{spike}}W_Q)), \quad K = \text{SN}(\text{BN}(X_{\text{spike}}W_K)), \quad V = \text{SN}(\text{BN}(X_{\text{spike}}W_V)). \tag{9}$$

Table 1: Performance comparison across datasets. Scores are reported as mAP for AudioSet (20K) and accuracy (%) for ESC-50 and SCV2. We evaluated three model variants: DiceFormer-10-S, DiceFormer-10-M, and DiceFormer-10-L. Here, "10" denotes a total of 10 attention layers, consisting of 5 SADA and 5 SDA layers each. The S, M, and L suffixes distinguish the models by their attention channel dimensions, which are set to [192, 384] for S, [256, 512] for M, and [384, 768] for L, respectively.

| Model | Type | Direct training | Parameters (M) | Energy (mJ) | Time step | Score |
|---|---|---|---|---|---|---|
| **AudioSet (20K)** — mAP | | | | | | |
| AST (Gong et al., 2021) | ANN | - | 88.1 | 475.64 | - | 0.148 / 0.347* |
| SSAST-S (Gong et al., 2022) | ANN | - | 23 | 176.82 | - | 0.165 / 0.308** |
| DTF-AT (Alex et al., 2024) | ANN | - | 69 | 153.18 | - | 0.187 / 0.355* |
| Spikformer (Zhou et al., 2023) | SNN | ✓ | 65.9 | 18.82 | 4 | 0.136 |
| Spike-driven Transformer (Yao et al., 2023) | SNN | ✓ | 65.9 | 8.15 | 4 | 0.130 |
| Spike-driven Transformer V2 (Yao et al., 2024) | SNN | ✓ | 55.0 | 14.92 | 4 | 0.117 |
| QKFormer (Zhou et al., 2024) | SNN | ✓ | 64.5 | 43.43 | 4 | 0.147 |
| DiceFormer-10-S (Ours) | SNN | ✓ | 13.7 | 5.34 | 4 | 0.145 |
| DiceFormer-10-M (Ours) | SNN | ✓ | 24.2 | 9.55 | 4 | 0.157 |
| DiceFormer-10-L (Ours) | SNN | ✓ | 54.3 | 6.18 | 1 | 0.153 |
| DiceFormer-10-L (Ours) | SNN | ✓ | 54.3 | 17.80 | 4 | **0.161** |
| **ESC-50** — Acc (%) | | | | | | |
| AST (Gong et al., 2021) | ANN | - | 87.2 | 260.54 | - | - / 88.7* |
| SSAST-S (Gong et al., 2022) | ANN | - | 23 | 72.63 | - | - /85.4** |
| DTF-AT (Alex et al., 2024) | ANN | - | 68.6 | 77.28 | - | 76.40 / 89.19* |
| DiceFormer-10-S (Ours) | SNN | ✓ | 13.5 | 5.17 | 4 | 85.37 |
| DiceFormer-10-M (Ours) | SNN | ✓ | 24.0 | 7.95 | 4 | **85.47** |
| **SCV2** — Acc (%) | | | | | | |
| AST (Gong et al., 2021) | ANN | - | 86.9 | 44.11 | - | - / 98.11* |
| SSAST-S (Gong et al., 2022) | ANN | - | 23 | 11.32 | - | 93.30 / 97.70** |
| DTF-AT (Alex et al., 2024) | ANN | - | 68.6 | 19.32 | - | 97.87 / 98.30* |
| DCLS-Delays (Hammouamri et al., 2024) | SNN | ✓ | 2.5 | - | - | 95.35 |
| SIDC-KWS (Lim & Kim, 2025a) | SNN | ✓ | 0.4 | - | 8 | 94.70 |
| DiceFormer-10-S (Ours) | SNN | ✓ | 13.5 | 5.31 | 4 | **97.27** |

\* ImageNet-pretrained;  \*\* Audio self-supervised pretraining.   a / b$^{(*,**)}$ = from-scratch / pretrained.

Attention is computed using the SDS. The resulting similarity map is passed through an SN to yield a binary spike-dice map, which is then applied to the value vector via element-wise Hadamard product with broadcasting:

$$\text{Attn} = \text{SN}(\text{SDS}(Q, K)) \odot V. \tag{10}$$

Finally, $\text{Attn}$ is reshaped back to the spatial view, passed through a Pointwise Convolution (PW-Conv) with BatchNorm, combined with the residual input $X_{\text{in}}$, and processed by an SN and an MLP to produce the final output:

$$X_{\text{fused}} = \text{BN}(\text{PWConv}(\text{Attn})) + X_{\text{in}}, \tag{11}$$
$$X_{\text{out}} = \text{MLP}(\text{SN}(X_{\text{fused}})). \tag{12}$$

Both SADA and SDA are naturally extended to multi-head variants, with detailed formulations provided in Appendix D.

## 4 EXPERIMENTS

We conduct extensive experiments to evaluate the effectiveness of DiceFormer across three widely used benchmark datasets: AudioSet (20K) and ESC-50 for general audio classification, and Speech Commands V2 (SCV2) for keyword spotting. These datasets collectively cover large-scale, environmental, and speech-focused tasks, providing a comprehensive assessment of model generalizability. All DiceFormer variants are trained from scratch to ensure that performance gains cannot be attributed to pre-training or external supervision. Our SNN models, in particular, adopt direct training (Wu et al., 2019) with surrogate gradients (Neftci et al., 2019; Fang et al., 2021). Detailed training protocols and hyperparameters are provided in Appendix E, while the energy calculation methodology and training/inference times are detailed in Appendix G and Appendix H, respectively.

### 4.1 PERFORMANCE COMPARISON

As presented in Table 1, on the AudioSet (20K) benchmark, our largest model, **DiceFormer-10-L** (54.3M, 17.80 mJ), achieves a new SNN SOTA with 0.161 mAP, trained entirely from scratch. It surpasses prior SNN-ViT models (Spikformer, Spike-driven Transformer, Spike-driven Transformer V2, QKFormer) while using fewer parameters and lower energy. Furthermore, our model outperforms the ANN baseline AST (88.1M, 475.64 mJ, 0.148 mAP) with substantially higher efficiency and accuracy. Although it remains 2.6 percentage points (pp) behind the ANN SOTA DTF-AT (69M, 153.18 mJ, 0.187 mAP), DiceFormer-10-L demonstrates markedly superior parameter and energy efficiency, highlighting the competitiveness of spiking architectures at scale.

Notably, the lightweight **DiceFormer-10-S** achieves **0.145 mAP** with only **13.7M** parameters and **5.34 mJ**, essentially matching the previous SNN SOTA QKFormer (64.5M, 43.43 mJ, 0.147 mAP), while using $\sim 4.7\times$ fewer parameters and $\sim 8.1\times$ lower energy. This highlights the scalability and efficiency of our design, even in resource-constrained settings.

Furthermore, on **ESC-50**, **DiceFormer-10-S** (13.5M, 5.17 mJ) reaches 85.37% accuracy, outperforming the from-scratch DTF-AT baseline (68.6M, 77.28 mJ, 76.40%) by a large margin while being far more efficient. On **SCV2**, **DiceFormer-10-S** (13.5M, 5.31 mJ) achieves 97.27%, competitive with the self-supervised-pretrained SSAST-S (23M, 11.32 mJ, 97.70%**) but with $\sim 2.1\times$ lower energy and $\sim 41\%$ fewer parameters.

Overall, as summarized in Table 1, these results demonstrate that DiceFormer not only establishes new SOTA performance among SNNs but also rivals strong ANN baselines, proving that spiking transformers can deliver both accuracy and efficiency across diverse audio tasks.

### 4.2 CORRELATION BETWEEN ATTENTION SCORES AND SPIKE DENSITY

On AudioSet-20K, we measure *density bias* as the Pearson correlation (Pearson, 1896) between layer-wise attention scores and input spike density (see Appendix F for details), conventional SNN-ViTs exhibit strong positive correlations, for example, Spikformer (0.934), Spike-driven Transformer (0.572), and QK-Former (0.725), indicating that their attention scores are heavily influenced by spike density. In contrast, DiceFormer substantially reduces this correlation, with values of 0.228 (10-S), 0.266 (10-M), and 0.169 (10-L). Across models, these correlations show a negative relationship with mAP (Pearson $r \approx -0.682$; Fig. 4), suggesting that lower density-dependence is a reliable indicator of better performance, even though other factors also contribute.

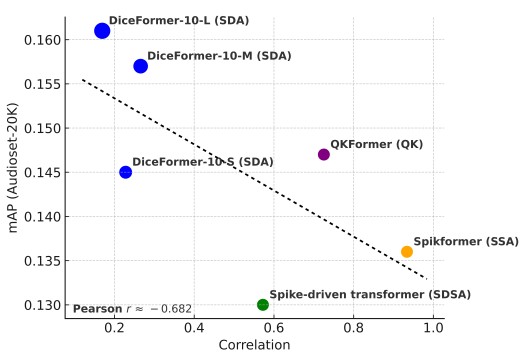

Figure 4: Association between mAP and density–score correlation.

### 4.3 ABLATION STUDY

All ablations are conducted on **AudioSet (20K)** with **DiceFormer-10-L** under a shared setup ($T=4$); results are summarized in Table 2. We analyze three aspects: (i) *component analysis* to validate the contribution of attention modules, **SADA** and **SDA**, in comparison to two density-unaware controls (dot-product **SSA** and Hadamard-product **SDSA**); (ii) the *generality* of **SDA** when used as a drop-in replacement in other SNNs; and (iii) the *effect of binarization* in enforcing a fully spike-based attention path.

**Analysis of attention components.** On AudioSet (20K), the full **SADA+SDA** model achieves **0.161** mAP. Using a single module yields 0.157 (SADA-only) and 0.158 (SDA-only), indicating that the two modules are complementary when combined. In contrast, density-unaware controls, SSA-only (dot-product, 0.142) and SDSA-only (Hadamard, 0.126), fall far behind. Relative to these baselines, SADA+SDA improves by +0.004 over SADA-only, +0.003 over SDA-only, +0.019 over SSA-only, and +0.035 over SDSA-only, under identical training ($T=4$) and comparable parameter

Table 2: Combined ablation studies on the AudioSet-20K dataset. The top section evaluates the effectiveness of SDA by swapping it into existing SNNs. The bottom sections analyze the synergy of attention components and the impact of attention map binarization.

| Methods | Parameters (M) | Time step | mAP |
|---|---|---|---|
| *Analysis of Attention Components* | | | |
| DiceFormer-10-L (SDSA only) | 55.4 | 4 | 0.126 |
| DiceFormer-10-L (SSA only) | 55.4 | 4 | 0.142 |
| DiceFormer-10-L (SADA only) | 54.3 | 4 | 0.157 |
| DiceFormer-10-L (SDA only) | 54.4 | 4 | 0.158 |
| DiceFormer-10-L (SADA + SDA, Full Model) | 54.3 | 4 | **0.161** |
| *Effectiveness of SDA* | | | |
| Spike-driven Transformer (baseline) | 65.9 | 4 | 0.130 |
| Spike-driven Transformer (+SDA) | 65.9 | 4 | **0.142 (+0.012)** |
| Spike-driven Transformer V2 (baseline) | 55.0 | 4 | 0.117 |
| Spike-driven Transformer V2 (+SDA)$^{\dagger}$ | 45.0 | 4 | **0.144 (+0.027)** |
| QKFormer (baseline) | 64.5 | 4 | 0.147 |
| QKFormer (+SDA) | 64.5 | 4 | **0.149 (+0.002)** |
| *Effectiveness of Binarized Attention Map* | | | |
| DiceFormer-10-L (SDS-continuous) | 54.3 | 4 | 0.160 |
| DiceFormer-10-L (SDS-binarized) | 54.3 | 4 | **0.161** |

$^{\dagger}$ Removes re-parameterized conv from attention block ($\approx$ -10M params).

counts ($\sim$ 54.2–55.4M). This confirms that density-aware scoring provides substantial advantages over conventional formulations.

**Effectiveness of SDA.** To isolate the impact of our core mechanism, we minimally swap **SDA** into three existing SNNs. In all cases, SDA yields consistent gains: Spike-driven Transformer $\rightarrow$ **0.130** $\rightarrow$ **0.142** (+0.012), Spike-driven Transformer V2 $\rightarrow$ **0.117** $\rightarrow$ **0.144** (+0.027), and QKFormer $\rightarrow$ **0.147** $\rightarrow$ **0.149** (+0.002). These results demonstrate that density-aware scoring generalizes across architectures and can serve as a reliable drop-in attention mechanism for SNNs.

**Effect of binarized SDS.** Finally, comparing the continuous (SDS-continuous, **0.160**) with its binarized variant (SDS-binarized, **0.161**) reveals negligible differences. This demonstrates that binarization preserves accuracy while enabling a purely spike-based attention computation, which is fully aligned with the event-driven nature of SNNs.

## 5 CONCLUSION

In this work, we proposed DiceFormer, a novel spiking transformer architecture tailored for audio processing. DiceFormer mitigates the density bias of conventional spike-based attention through SDA, a linear-complexity mechanism that yields density-aware attention scores. To further capture audio-specific patterns, we introduced SADA, which decouples and fuses frequency-temporal structure. Ablation studies confirmed the complementary contributions of SDA and SADA, the robustness of their binarized realization, and the generality of SDA as a drop-in replacement across SNNS. Trained entirely from scratch, DiceFormer-10-L achieves 0.161 mAP on AudioSet (20K), establishing a new SOTA performance among SNNs, and delivers competitive or superior results on ESC-50 and SCV2 with strong parameter and energy efficiency.

**Limitations and Future Work.** Our experiments trained DiceFormer from scratch to isolate architectural contributions and efficiency. This design choice currently leaves a gap compared to state-of-the-art ANNs that leverage large-scale pretraining. As future work, we will explore spike-compatible pretraining and distillation strategies—such as self-supervised audio pretraining and multimodal teacher–student transfer—built upon our SDA/SADA modules. We believe these directions hold strong potential to further narrow the performance gap and advance the practical adoption of SNNs in real-world audio applications.

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

## A    SPIKING NEURAL NETWORK

Spiking Neural Networks (SNNs), inspired by the information processing in the human brain, operate in an event-driven manner using sparse, binary spikes. This bio-mimetic approach offers significant potential for energy efficiency. However, a primary challenge in training SNNs is the non-differentiable nature of the spike generation function within neuron models, such as the Leaky Integrate-and-Fire (LIF) neuron (Maass, 1997; Wu et al., 2017), which precludes the direct application of standard backpropagation (Rumelhart et al., 1986). To address this, two main training paradigms have emerged. The first is ANN-to-SNN conversion (Deng & Gu, 2021; Hu et al., 2023; Han et al., 2020; Cao et al., 2015; Wang et al., 2022), which transfers the weights from a pre-trained ANN to an SNN architecture. The second is direct training (Wu et al., 2019), which employs the surrogate gradient (Neftci et al., 2019) method to approximate the derivative of the spike activation. In this work, we utilize the direct training paradigm to train our proposed model.

## B    DICEFORMER IMPLEMENTATION DETAILS

We configure three versions of the DiceFormer model based on the attention channel dimensions: DiceFormer-10-S, DiceFormer-10-M, and DiceFormer-10-L. The number 10 in the model names denotes the total count of SADA and SDA blocks, with all versions containing a total of 10 such blocks. The detailed architecture for each model is summarized in **Table 3**.

The model processes an input tensor with a shape of $T \times C \times H \times W$, where $T, C, H,$ and $W$ denote the time step, channels, height, and width, respectively. For simplicity, the batch dimension is omitted from all tensor shapes described below.

Table 3: Detailed architecture of DiceFormer. 'k', 's', and 'p' denote kernel size, stride, and padding, respectively. The model has three variants: **DiceFormer-10-S** ($E = 96, D_1 = 192, D_2 = 384$), **DiceFormer-10-M** ($E = 128, D_1 = 256, D_2 = 512$), and **DiceFormer-10-L** ($E = 192, D_1 = 384, D_2 = 768$). For all variants, the SADA and SDA modules use 2 and 8 attention heads, respectively.

| Stage | Layer Name | Key Operations | # Tokens | Channels | | |
|---|---|---|---|---|---|---|
| | | | | DiceFormer-10-S | DiceFormer-10-M | DiceFormer-10-L |
| 1 | Initialize Block | Init-1:
  Conv2d(k=7,s=1,p=3)
  MaxPool(k=2,s=2)
Init-2:
  Conv2d(k=3,s=1,p=1)
  MaxPool(k=2,s=2)
Init-3:
  Conv2d(k=3,s=1,p=1)
Skip:
  Conv2d(k=3,s=2,p=1) | $\frac{H}{4} \times \frac{W}{4}$ | 96 | 128 | 192 |
| 2 | Projection Conv 1 | Project-1:
  Conv2d(k=3,s=1,p=1)
  MaxPool(k=2,s=2)
Project-2:
  Conv2d(k=3,s=1,p=1) | $\frac{H}{8} \times \frac{W}{8}$ | 192 | 256 | 384 |
| 3 | SADA Module | – *Frequency Stream* –
Freq-Attention (2 Heads) | $\frac{H}{8} \times \frac{W}{8}$ | 96 | 128 | 192 |
| | | – *Temporal Stream* –
Temp-Attention (2 Heads) | $\frac{H}{8} \times \frac{W}{8}$ | 96 | 128 | 192 |
| | | PWConv & MLP | $\frac{H}{8} \times \frac{W}{8}$ | 192 | 256 | 384 |
| 4 | Projection Conv 2 | Project-1:
  Conv2d(k=3,s=1,p=1)
  MaxPool(k=2,s=2)
Project-2:
  Conv2d(k=3,s=1,p=1) | $\frac{H}{16} \times \frac{W}{16}$ | 384 | 512 | 768 |
| 5 | SDA Module | Unified Spike Dice Attention (8 Heads) | $\frac{H}{16} \times \frac{W}{16}$ | 384 | 512 | 768 |
| | | Fusion (PWConv) & MLP | $\frac{H}{16} \times \frac{W}{16}$ | 384 | 512 | 768 |
| 6 | Classification Head | Global Average Pooling, Linear | – | | - | |

## C    SPIKE DICE ATTENTION: THEORETICAL PROPERTIES

This appendix provides a formal proof that the proposed SDA is density-aware. The analysis matches the main-paper definition: the SDS is computed per token along the channel dimension with $\epsilon > 0$, yielding $\text{SDS}(Q, K) \in \mathbb{R}^{N \times 1}$ and overall $O(ND)$ complexity.

**Setup and notation.** Let $Q, K \in \{0, 1\}^{N \times D}$. For the $n$-th token we write $q_n = (q_{n,1}, \ldots, q_{n,D}) \in \{0, 1\}^D$ and $k_n = (k_{n,1}, \ldots, k_{n,D}) \in \{0, 1\}^D$. Define the (spike) densities $\|q_n\|_1 = \sum_{d=1}^{D} q_{n,d}$ and $\|k_n\|_1 = \sum_{d=1}^{D} k_{n,d}$, and the co-occurrence $q_n \cdot k_n = \sum_{d=1}^{D} q_{n,d} k_{n,d}$. The per-token SDS equals

$$s_n = \frac{2 (q_n \cdot k_n)}{\|q_n\|_1 + \|k_n\|_1 + \epsilon}, \qquad \epsilon > 0. \tag{13}$$

Stacking $\{s_n\}_{n=1}^{N}$ gives $\text{SDS}(Q, K) \in \mathbb{R}^{N \times 1}$. Because each token only aggregates over channels $D$, the total cost is $O(ND)$.

**Limitations of density-unaware raw scores.** For binary vectors, the raw dot/Hadamard score $\text{Score}_{\text{raw}}(n) := q_n \cdot k_n$ cannot distinguish keys that share the same co-occurrence and is biased toward saturated (all-ones) keys: if $k = \|q_n\|_1 \geq j$, then $q_n \cdot k_{\text{sat}} = k \geq j$.

### C.1 PREFERENCE FOR SPARSE KEYS AT EQUAL CO-OCCURRENCE

Fix $q_n$ and consider two keys with equal overlap $q_n \cdot k_{\text{sparse}} = q_n \cdot k_{\text{dense}} = j$ but different densities $\|k_{\text{sparse}}\|_1 = j$ and $\|k_{\text{dense}}\|_1 = m > j$. Then

$$s_n(k_{\text{sparse}}) = \frac{2j}{\|q_n\|_1 + j + \epsilon}, \qquad s_n(k_{\text{dense}}) = \frac{2j}{\|q_n\|_1 + m + \epsilon}. \tag{14}$$

Since $\|q_n\|_1 + m + \epsilon > \|q_n\|_1 + j + \epsilon$, we have

$$s_n(k_{\text{sparse}}) > s_n(k_{\text{dense}}). \tag{15}$$

Hence, SDA assigns a higher score to lower-density keys at equal co-occurrence.

### C.2 SUPPRESSION OF SATURATED KEYS

For the saturated (all-ones) key $k_{\text{sat}}$ we have $\|k_{\text{sat}}\|_1 = D$ and $q_n \cdot k_{\text{sat}} = \|q_n\|_1 =: k$. Equation equation 13 gives

$$s_n(k_{\text{sat}}) = \frac{2k}{k + D + \epsilon} \xrightarrow{D \to \infty} 0. \tag{16}$$

In contrast, an ideal sparse key $k_{\text{sparse}}$ maintains $s_n(k_{\text{sparse}}) = \frac{2j}{k+j+\epsilon}$, which is independent of $D$.

## D    SPIKE DICE ATTENTION: MULTI-HEAD FORMULATION

### (1) MULTI-HEAD IN SADA

The SADA module applies multi-head attention to both the frequency and temporal streams. In each stream, the input tensor has shape $T \times N \times D$ (with stream-specific $N$ and $D$). We split the feature dimension $D$ into $i$ heads, each with a per-head width of $d = D/i$, and reshape it to $T \times i \times N \times d$, enabling parallel attention across heads.

**Frequency branch (multi-head).** Given $Q_f, K_f, V_f \in \{0, 1\}^{T \times N_{\text{freq}} \times D_1'}$ with $D_1' = id$, for each head $h \in \{1, \ldots, i\}$ we take $Q_f^{(h)}, K_f^{(h)}, V_f^{(h)} \in \{0, 1\}^{T \times N_{\text{freq}} \times d}$ and compute

$$\text{Attn}_{\text{freq}}^{(h)} = \text{SN}(\text{SDS}(Q_f^{(h)}, K_f^{(h)})) \odot V_f^{(h)}.$$

The final frequency output is obtained by merging head-wise results along the head axis to recover the original width: $\text{Attn}_{\text{freq}} \in \{0, 1\}^{T \times N_{\text{freq}} \times D_1'}$.

**Temporal branch (multi-head).** Analogously, with $Q_t, K_t, V_t \in \{0,1\}^{T \times N_{\text{temp}} \times D_1'}$ and $D_1' = id$, we compute

$$\text{Attn}_{\text{temp}}^{(h)} = \text{SN}(\text{SDS}(Q_t^{(h)}, K_t^{(h)})) \odot V_t^{(h)},$$

and merge along the head axis to obtain $\text{Attn}_{\text{temp}} \in \{0,1\}^{T \times N_{\text{temp}} \times D_1'}$.

3. FEATURE FUSION AND OUTPUT.

The outputs of the two streams, $\text{Attn}_{\text{freq}}$ and $\text{Attn}_{\text{temp}}$, are concatenated along the channel dimension, restoring the original channel $D_1$. The combined tensor is passed through a PWConv to fuse inter-dependencies, followed by a residual connection and an MLP:

$$X_{\text{fused}} = \text{BN}(\text{PWConv}(\text{concat}(\text{Attn}_{\text{freq}}, \text{Attn}_{\text{temp}}))) + X_{\text{in}},$$

$$X_{\text{out}} = \text{MLP}(\text{SN}(X_{\text{fused}})),$$

(2) MULTI-HEAD IN SDA (UNIFIED)

The unified SDA also employs multi-head attention. The input tensor $T \times N_{HW} \times D_2$ is split into $i$ heads, yielding $d = D_2/i$ and the reshaped form $T \times i \times N_{HW} \times d$ for parallel processing.

**Unified branch (multi-head).** For $Q, K, V \in \{0,1\}^{T \times N_{HW} \times D_2}$ with $D_2 = id$ and $N_{HW} = (H/16)(W/16)$, each head computes

$$\text{Attn}^{(h)} = \text{SN}(\text{SDS}(Q^{(h)}, K^{(h)})) \odot V^{(h)}.$$

Merging the $i$ head-wise outputs restores the width $D_2$: $\text{Attn} \in \{0,1\}^{T \times N_{HW} \times D_2}$.

**Residual + PWConv.**

$$X_{\text{fused}} = \text{BN}(\text{PWConv}(\text{Attn})) + X_{\text{in}},$$

$$X_{\text{out}} = \text{MLP}(\text{SN}(X_{\text{fused}})).$$

# E  EXPERIMENT DETAILS

## E.1  TRAINING ENVIRONMENT

For our experiments, we trained the models for AudioSet (20K) on a single NVIDIA A100 GPU with 80 GB of memory. For the ESC-50 and SCV2 datasets, training was conducted on a single NVIDIA RTX 6000 GPU with 96GB of memory.

## E.2  SURROGATE GRADIENT FUNCTION

To address the non-differentiable nature of the spike activation function in direct training, we employed the surrogate gradient method. Specifically, we adopted the sigmoid function as our surrogate function, which is defined as:

$$\text{Sigmoid}(x) = \frac{1}{1 + \exp(-\gamma x)} \tag{17}$$

The steepness parameter $\gamma$ was set to 4.0 in our experiments.

## E.3  DATASETS

We employed Specaugment (Park et al., 2019) and Mixup (Zhang et al., 2017) during training. Detailed experimental settings for each dataset are summarized in Table 4.

**AudioSet.** AudioSet is a large-scale dataset for audio event research, containing 10-second audio clips sourced from YouTube videos. The dataset is organized with an ontology of 527 audio event classes (Gemmeke et al., 2017). It includes a smaller *balanced* subset ( 22k clips) for controlled experiments and an *evaluation* set ( 20k clips). For our work, we utilize this dataset as a foundational resource for training general-purpose audio representations. For our experiments, all audio clips were first resampled to 16 kHz and converted to a single channel (mono). We then extracted log-Mel filterbank energies (Fbank) as input features. The features were computed using a Hanning window with a frame shift of 10 ms. We used 128 Mel frequency bins for the filterbank calculation. To ensure a uniform input size for our model, the resulting feature sequences were padded or truncated to a fixed length of 1024 frames.

**ESC-50 (Environmental Sound Classification).** The ESC-50 dataset is a widely used benchmark for environmental sound classification (Piczak, 2015). It consists of 2,000 5-second audio recordings distributed evenly across 50 distinct semantic classes, such as "Dog" and "Rain," with 40 clips per class. Following standard evaluation protocol, we report the classification accuracy using a 5-fold cross-validation scheme. The audio data from ESC-50 was processed using the same feature extraction methodology applied to AudioSet. After converting the recordings into log-Mel filterbank features, each sequence was padded to a final length of 512 frames.

**Speech Commands V2 (SCV2).** The Speech Commands V2 dataset is designed for keyword spotting tasks (Warden, 2018). It contains approximately 105,000 1-second utterances of 35 short command words. The dataset is pre-divided into standard training, validation, and testing splits. Performance is evaluated based on classification accuracy on the test set. Following the same procedure, we extracted Fbank features from each utterance. The temporal dimension of the resulting feature sequences was then padded to 128 frames to create uniform inputs for our model.

Table 4: Experiment details for different datasets.

| Hyperparameter | Audioset | ESC-50 | SCV2 |
|---|---|---|---|
| Time step | 4 | 4 | 4 |
| Batch size | 12 | 24 | 96 |
| Optimizer | AdamW | AdamW | AdamW |
| Input shape | (1, 1024, 128) | (1, 512, 128) | (1, 128, 128) |
| **Augmentation** | spec | spec + Mixup | spec + Mixup |
| Epochs | 100 | 200 | 200 |
| Scheduler | Cosine | Cosine | Cosine |
| Warmup epochs | 5 | 5 | 5 |
| Warmup start lr | $1 \times 10^{-3}$ | $1 \times 10^{-4}$ | $1 \times 10^{-4}$ |
| Warmup end lr | $1 \times 10^{-2}$ | $1 \times 10^{-3}$ | $1 \times 10^{-3}$ |
| End lr | $5 \times 10^{-5}$ | $1 \times 10^{-6}$ | $1 \times 10^{-6}$ |

## F    CORRELATION ANALYSIS DETAILS

Density–unaware spike attention can exhibit a high correlation between the attention score and the spike density, implying that scores may be biased by how many spikes fire rather than by semantic similarity. This limitation undermines the reliability of attention scores. In this section, we quantify such density bias on AudioSet-20K for Spikformer (SSA), Spike-driven Transformer (SDSA), DiceFormer-10-{S,M,L} (SDA), and QKFormer (QK) by measuring correlations between layer-wise attention scores and the spike densities of their inputs across the available attention layers (five for Spikformer/SDSA/DiceFormer; three for QKFormer).

**Spike density.** At each layer, we define "spike density" as the fraction of active spikes in the tensors (e.g., $Q/K$ or $Q/K/V$, depending on the mechanism) that are actually used to compute the attention score.

**Results.** As detailed in Table 5, the correlations averaged over the available stages are highest for Spikformer (0.934, range [0.920, 0.949]) and moderate for the Spike-driven Transformer (0.572, [0.545, 0.595]). For DiceFormer-10-{S,M,L}, the means (ranges) are 0.228 ([−0.061, 0.349]), 0.264 ([0.113, 0.368]), and 0.196 ([−0.0038, 0.409]), respectively; QKFormer shows 0.725 ([0.677, 0.766]) over three attention layers. These results indicate that our density-aware formulation mitigates spike–density bias more effectively than dot-product/Hadamard-style attention.

Table 5: Density correlation analysis on AudioSet-20K. Correlation between attention score and spike density by stage (where available); mean and range are computed across the reported attention layers.

| Model | Stage 0 | Stage 1 | Stage 2 | Stage 3 | Stage 4 | Mean | Range |
|---|---|---|---|---|---|---|---|
| Spikformer (SSA) | 0.949 | 0.930 | 0.935 | 0.934 | 0.920 | 0.934 | [0.920, 0.949] |
| Spike-driven transformer (SDSA) | 0.565 | 0.585 | 0.595 | 0.571 | 0.545 | 0.572 | [0.545, 0.595] |
| QKFormer (QK)[†] | 0.678 | 0.732 | 0.767 | — | — | 0.725 | [0.678, 0.767] |
| DiceFormer-10-S (SDA) | 0.277 | 0.228 | -0.061 | 0.349 | 0.347 | 0.228 | [-0.061, 0.349] |
| DiceFormer-10-M (SDA) | 0.264 | 0.368 | 0.319 | 0.113 | 0.264 | 0.266 | [0.113, 0.368] |
| DiceFormer-10-L (SDA) | 0.196 | 0.238 | -0.038 | 0.042 | 0.409 | 0.169 | [-0.038, 0.409] |

[†] QKFormer applies QK attention in only three stages (Stages 0–2); mean/range are computed over these available stages.

## G   Energy Consumption Estimation Methodology

We estimate the theoretical energy consumption based on the total Synaptic Operations (SOPs) (Kundu et al., 2021; Hu et al., 2021; Yao et al., 2022; Zhou et al., 2024; 2023), which represent spike-based accumulate (AC) operations, calculated from the FLOPs (floating point operations) of each layer. We distinguish between continuous-valued inputs (MAC operations) and spike-based inputs (AC operations).

The theoretical FLOPs for each layer are calculated according to standard definitions (Molchanov et al., 2017) as follows:

- **Conv2d:** FLOPs $= C_{\text{out}} \times (C_{\text{in}}/g) \times k_h k_w \times H_{\text{out}} W_{\text{out}}$
- **Conv1d:** FLOPs $= C_{\text{out}} \times (C_{\text{in}}/g) \times k \times L_{\text{out}}$
- **Linear:** FLOPs $= \text{in\_features} \times \text{out\_features}$

We adopt the energy constants from a 45nm CMOS process reported in (Horowitz, 2014):

$$E_{\text{MAC}} = 4.6 \, \text{pJ} \tag{18}$$
$$E_{\text{AC}} = 0.9 \, \text{pJ} \tag{19}$$

1. **SNN Energy :** For spike-driven layers, we scale by time steps $T$ and the layer-wise spike rate $R_\ell \in [0, 1]$, using the energy per accumulate:

$$\text{SOPs}_\ell = \text{FLOPs}_\ell \times T \times R_\ell,$$
$$E_{\text{SNN}} = \sum_\ell \text{SOPs}_\ell \times E_{\text{AC}}. \tag{20}$$

We refer to $\text{FLOPs}_\ell \times T \times R_\ell$ as the spiking operation count (SOPs) of layer $\ell$.

2. **Total Energy:** The total energy is obtained by summing the per-layer SNN energies defined above.

## H   Training and Inference Time

The DiceFormer-10-L, DiceFormer-10-M, DiceFormer-10-S, Spikformer, Spike-driven transformer, Spike-driven transformer V2 and QKFormer models were trained and evaluated on the AudioSet (20K) dataset. The actual training and inference times, measured on the same device with a batch size (BS) of 12, are detailed in Table 6.

Table 6: Training and inference time per epoch for various models on the AudioSet (20K) dataset. All measurements were performed on the same device.

| Model | Training Time (min sec / epoch) | Inference Time (min sec / epoch) |
|---|---|---|
| DiceFormer-10-S | 6m 43s | 2m 32s |
| DiceFormer-10-M | 8m 39s | 3m 23s |
| DiceFormer-10-L | 13m 45s | 5m 16s |
| Spikformer | 9m 33s | 3m 32s |
| Spike-driven transformer | 9m 15s | 3m 26s |
| Spike-driven transformer V2 | 12m58s | 4m 15s |
| QKFormer | 12m 06s | 4m 29s |

## I LARGE LANGUAGE MODEL USAGE

We received assistance from the GPT-5-Thinking and Gemini 2.5 Pro models in refining the grammar and style of this manuscript.