# OpenReview forum: "DiceFormer: Spiking Audio Transformer with Density-Aware Dice Attention"
_ICLR.cc/2026/Conference — Submitted to ICLR 2026_

### Official Review · Reviewer_T9Rj · 2025-10-28

**Soundness:** 2
**Presentation:** 2
**Contribution:** 2
**Rating:** 4
**Confidence:** 4

**Summary:**

The paper proposes DiceFormer, an SNN transformer for audio with two key modules: SDA (Spike Dice Attention), a density-aware, linear-time attention based on a Dice-style score, and SADA (Spike Audio Dice Attention), a frequency/temporal dual-branch variant for spectrograms. It reports strong from-scratch results on AudioSet-20K (mAP 0.161 with DiceFormer-10-L), ESC-50, and Speech Commands V2, and shows SDA can be dropped into prior SNN ViTs for gains.

**Strengths:**

1) The paper diagnoses spike-density bias in dot-product/Hadamard attention and introduces a Dice-based score to normalize by token activity, keeping O(ND) complexity.

2) DiceFormer-10-L reaches 0.161 mAP on AudioSet-20K and shows favorable parameter/energy figures compared with SNN/ANN baselines; smaller variants are also strong.

3) Replacing attention in prior SNN ViTs with SDA improves their scores, suggesting the mechanism’s usefulness beyond this architecture.

**Weaknesses:**

1) The paper lists SDA and SADA as two main contributions, but SADA is essentially a dual-branch time–frequency decoupling—a strategy already explored by DTF-AT (AAAI’24)，yet the contribution currently reads as incremental in architecture (single- vs. dual-branch) rather than conceptually new.
2) Prior SNN Transformers (e.g., Spikformer, Spike-driven Transformer, QKFormer) emphasize avoiding MACs in favor of AC-style primitives for spiking self-attention. In contrast, Eq. (4) introduces an input-dependent division (normalizing by the sum of activities), which is not offline-precomputable and is often **unfriendly to neuromorphic substrates** (division/reciprocal are costly, latency-prone, and may break integer/accumulate-only pipelines). So, my major concern is that such division-based operations fundamentally limit neuromorphic deployability.  The paper should quantify and justify this choice.
3) DiceFormer uses **PLIF** neurons (learnable τ). It’s not specified whether compared SNN baselines in the authors’ re-implementations also use PLIF (or just LIF). If DiceFormer benefits from PLIF while baselines use LIF, that could confound fairness.
4) The paper standardizes “10 layers = 5 SADA + 5 SDA” across S/M/L but it does not examine alternative SADA:SDA **ratios** (eg., 4:6 or 6:4) nor different **orderings** (e.g., front-loaded SADA vs. interleaved vs. back-loaded). Such ablations are important to verify complementarities and to identify where each module provides the largest gains.

**Questions:**

Please refer to the four points included in **Weaknesses** for details.

---

> ### Author Response · Authors · 2025-11-19
>
> We appreciate Reviewer T9Rj time and constructive feedback. Thanks to your comments, we were able to clarify key aspects of our paper.
>
> > Weakness-1
> > The paper lists SDA and SADA as two main contributions, but SADA is essentially a dual-branch time–frequency decoupling—a strategy already explored by DTF-AT (AAAI’24)，yet the contribution currently reads as incremental in architecture (single- vs. dual-branch) rather than conceptually new.
>
> As the reviewer correctly points out, the high-level idea of SADA—time–frequency dual-branch decoupling—has already been explored in prior ANN-based audio Transformers such as DTF-AT [1]. We fully agree with this observation, and in the camera-ready version we will carefully soften and adjust the description of our contributions so that SADA is not presented as a “completely new time–frequency decoupling paradigm.”
> However, the role and contribution of SADA in this paper is not to claim an entirely novel architectural concept per se, but rather to serve as a spike-based module that extends our main contribution, Spike Dice Attention (SDA), to the audio domain in a specialized way. We clarify this intent in more detail below.
>
> **1. Dual-branch structures are familiar in ANN audio, but this is the first application to SNN foundation models**
>
> The time–frequency dual-branch structure itself is a well-known design in ANN-based audio Transformers such as DTF-AT [1]. However, existing SNN foundation models [2–5] have been almost exclusively designed for the vision domain, and as a result:
> * They do not incorporate time–frequency module structures tailored to audio, and
> * They tend to reuse single-branch architectures originally designed for images directly on spectrograms.
>
> To the best of our knowledge, there has been no prior work that applies a dual-branch time–frequency structure to an SNN-based audio foundation model. Thus, the goal of SADA is not to claim that we “invented” the dual-branch concept itself, but rather to:
> take a well-established audio design principle from ANNs (dual-branch time–frequency decoupling),
> and redesign and validate it under spike-based SNN constraints (binary spikes, event-driven updates, sparsity, spike accumulation, etc.).
> From this perspective, SADA plays a role analogous to early SNN-Transformer works that did not invent the Vision Transformer itself, but redesigned ViT architectures into spike-former structures appropriate for SNNs. Those works are widely regarded as meaningful contributions in the SNN community, and we view SADA in a similar position for audio SNN foundation models.
>
> **2. The core conceptual contribution is SDA; SADA is its audio-domain specialization**
>
> The central conceptual contribution of this paper is Spike Dice Attention (SDA). SDA explicitly analyzes and addresses the spike density-unaware problem that is commonly present in current spike-based self-attention formulations.
> As discussed in Section 3.3 and Appendix C, existing methods such as dot-product SSA and Hadamard-based SDSA compute similarity purely based on spike counts (firing rates). Consequently:
> * A token may receive a very large attention score even if its true semantic similarity is not high,
> * Simply because its spike density is high.
>
> This behavior diverges from the intention of ANN attention—“focusing on truly important tokens”—and makes it difficult to faithfully reproduce ANN attention behavior on spike tensors.
> To resolve this, SDA introduces a Dice-based spike similarity (SDS) with spike density normalization, which:
> * Suppresses tokens that are merely high-density, and
> * Assigns higher scores to “sparse but accurate matches” instead.
>
> We mathematically analyze this behavior in Appendix C, and show that SDA yields attention scores that more closely approximate the functional behavior of ANN attention under SNN constraints.
> Empirically, as shown in Table 2, simply replacing the spike-based attention in existing SNN-Transformers with SDA under the same training and evaluation settings yields consistent performance improvements. For example:
> * Spike-driven Transformer [3]: +0.012 mAP
> * Spike-driven Transformer V2 [4]: +0.027 mAP
> * QKFormer [5]: +0.002 mAP
>
> These results demonstrate that SDA quantitatively alleviates the density-unaware limitation of prior spike-based attention mechanisms.
> In this context, SADA is positioned not as a standalone, fundamentally new conceptual module, but as an “audio-domain specialization” of SDA. Concretely:
> * The dual-branch structure itself follows existing ANN audio Transformers such as DTF-AT [1],
> * Each branch’s attention is replaced with density-aware SDA, and
> * The entire module is designed to operate under spike-based, event-driven constraints as an audio SNN-Transformer block.

---

> ### Author Response · Authors · 2025-11-19
>
> In short,
> * SADA ≠ a completely new time–frequency decoupling paradigm,
> * SADA = a practical instantiation that specializes our main contribution (SDA) to the audio SNN setting on top of a dual-branch time–frequency structure.
> This is the intended scope of our claim.
>
> **3. Why this adaptation is still meaningful in the SNN context**
>
> Even though dual-branch architectures are well known in ANN-based audio Transformers, their behavior and effectiveness are not automatically guaranteed when ported to SNNs. For example, it is not obvious:
> * Whether the dual-branch structure remains effective when the input is binary spikes,
> * How spike-density–driven score inflation manifests in each branch,
> * How time and frequency representations behave under SNN-specific constraints (sparsity, temporal accumulation, etc.), and
> * Whether SDA can operate stably within a dual-branch structure and still provide performance gains.
>
> Our experimental results show that:
> * SADA (dual-branch SDA) achieves higher mAP than both single-branch SDA and existing SNN-Transformers, and
> * Dual-branch time–frequency modeling is not only viable but can be advantageous in SNN foundation models.
>
> In other words, we are not merely copying a “known ANN design” and applying it as-is. Rather, we:
> combine a newly proposed spike-based attention (SDA) with a dual-branch structure,
> and perform, to our knowledge, the first systematic design and validation of a dual-branch audio SNN architecture that respects SNN constraints.
> We believe this constitutes a meaningful extension in the SNN context.
>
> **4. Planned revisions in the manuscript**
>
> Reflecting the reviewer’s comments, we will revise the manuscript as follows:
> * **In the contribution section:**
>     * We will explicitly state that the dual-branch time–frequency structure itself follows prior ANN audio Transformers such as DTF-AT [1].
>     * We will emphasize that the main conceptual contribution is the formulation and analysis of SDA as a spike density-aware self-attention mechanism.
> * **In the description of SADA:**
>     * We will avoid phrasing that suggests SADA is a “new architectural paradigm.”
>     * Instead, we will describe SADA as
>         * “A dual-branch SNN-Transformer module that applies SDA to the audio domain,” and
>         * “The first time–frequency dual-branch design applied to an audio SNN foundation model.”
>
> Through these clarifications, we aim to make it explicit that we are not over-claiming the novelty of SADA itself; the core conceptual contribution lies in SDA, while SADA is a domain-specific, empirically validated extension of SDA to the audio SNN setting.
>
> **References**
>
> [1] DTF-AT: Decoupled Time-Frequency Audio Transformer for Event Classification. In AAAI 2024.
>
> [2] Spikformer: When Spiking Neural Network Meets Transformer. ICLR 2023.
>
> [3] Spike-driven Transformer. NeurIPS 2023.
>
> [4] Spike-driven Transformer V2. ICLR 2024.
>
> [5] QKFormer: Hierarchical Spiking Transformer using Q-K Attention. NeurIPS 2024.

---

> ### Author Response · Authors · 2025-11-19
>
> > Weakness-2
> > Prior SNN Transformers (e.g., Spikformer, Spike-driven Transformer, QKFormer) emphasize avoiding MACs in favor of AC-style primitives for spiking self-attention. In contrast, Eq. (4) introduces an input-dependent division (normalizing by the sum of activities), which is not offline-precomputable and is often unfriendly to neuromorphic substrates (division/reciprocal are costly, latency-prone, and may break integer/accumulate-only pipelines). So, my major concern is that such division-based operations fundamentally limit neuromorphic deployability. The paper should quantify and justify this choice.
>
> **Response to Weakness-2**
>
> We sincerely thank the reviewer for raising this crucial question. We fully agree that maintaining neuromorphic friendliness is a paramount issue in SNN design.
>
> We have addressed this specific concern regarding the division operation and its hardware implementation in detail in our **Global Rebuttal: On the spike-driven and neuromorphic-friendly nature of SDA**. We kindly invite the reviewer to refer to that section.

---

> ### Author Response · Authors · 2025-11-19
>
> > Weakness-3
> > DiceFormer uses PLIF neurons (learnable $\\tau$). It’s not specified whether compared SNN baselines in the authors’ re-implementations also use PLIF (or just LIF). If DiceFormer benefits from PLIF while baselines use LIF, that could confound fairness.
>
> **Response to Weakness-3**
>
> We thank the reviewer for raising this important concern that the use of PLIF neurons could potentially affect the fairness of the comparative experiments. As correctly pointed out, the manuscript did not describe in sufficient detail which neuron types (LIF / PLIF / variant LIF) are used in each model. We clarify the implementation details and report additional experimental results below.
>
> **1. PLIF vs. LIF: Same neuron model, different way of setting $\\tau$**
>
> The LIF/PLIF neurons used in this work are based on the spikingjelly library [1]. The membrane potential update of the LIF neuron is defined as:
> $$
> H[t] = V[t-1] + \\frac{1}{\\tau}\\big( X[t] - (V[t-1] - V_{\\text{reset}}) \\big).
> $$
> Here, LIF and PLIF share exactly the same update equation, and the only difference lies in how the time constant ($\\tau$) is determined:
> * **LIF:** $\\tau$ is a fixed hyper-parameter, typically tuned manually via grid search or heuristic optimization.
> * **PLIF:** $\\tau$ is treated as a learnable parameter and is automatically adapted to the data through backpropagation.
>
> In other words, PLIF is not a “stronger or fundamentally different neuron model”; it is the same neuron model, differing only in whether $\\tau$ is manually tuned (LIF) or learned (PLIF).
> Our choice of PLIF in DiceFormer was therefore a practical design decision to avoid extensive hyper-parameter search over $\\tau$, rather than an attempt to “secretly” boost performance by using a more powerful neuron.
>
> **2. Neuron configurations of the compared SNN-Transformers**
>
> All SNN-Transformers re-implemented in this paper follow the official implementations and settings of their respective original works. The neuron types are summarized as follows:
> * Spikformer [2] – LIF
> * Spike-driven Transformer [3] – LIF
> * Spike-driven Transformer V2 [4] – LIF-family neuron with a learnable firing threshold
> * QKFormer [5] – LIF
>
> Thus, all baseline models use the LIF / threshold settings proposed in their original papers, and we did not modify their neuron structures in a way that would unfairly disadvantage them.
>
> **3. Direct fairness test: LIF-only DiceFormer (removing PLIF entirely)**
>
> To directly address the reviewer’s question of whether “DiceFormer gains an unfair advantage because it uses PLIF,” we conducted an additional experiment where all PLIF neurons in DiceFormer were replaced with LIF neurons.
> To ensure fairness, we:
> * Did not perform any new tuning or search over $\\tau$, and
> * Used a conservative setting where we simply keep the commonly used default initialization and hyper-parameters for $\\tau$.
>
> On AudioSet, the mAP results of this LIF-only DiceFormer are:
>
> | Model | mAP |
> | :--- | :--- |
> | DiceFormer-10-L (LIF) | 0.154 |
> | DiceFormer-10-M (LIF) | 0.151 |
> | DiceFormer-10-S (LIF) | 0.142 |
>
> Compared to the PLIF versions, the performance differences are only about 0.007 / 0.006 / 0.003 mAP, respectively, showing that the performance change is very limited even without any additional tuning of $\\tau$.
> More importantly, even this LIF-only DiceFormer still achieves higher mAP than existing SNN-Transformers [2–5].
> Therefore:
> * The main performance gains of DiceFormer are preserved even after completely removing PLIF,
> * The advantage of DiceFormer primarily comes from the SDA/SADA-based architectural design, and
> * PLIF serves as a “convenience choice” to automate $\\tau$ selection, rather than a decisive factor that compromises fairness.
>
> **4. Summary and planned revisions**
>
> In summary:
> 1.  PLIF and LIF are mathematically the same neuron model; the only difference is whether the time constant ($\\tau$) is chosen manually (LIF) or optimized through learning (PLIF).
> 2.  Even when we remove PLIF entirely and configure DiceFormer to use only LIF neurons, the performance drop is limited to about 0.003–0.007 mAP, and DiceFormer still outperforms all existing SNN-Transformers.
> 3.  All baseline models follow the LIF / threshold settings in their original papers, so there is no manipulation of neuron configurations that would systematically favor our method.
> 4.  Thus, we conclude that the primary performance improvements of DiceFormer stem from the proposed SDA/SADA-based architecture, not from the mere use of PLIF.

---

> ### Author Response · Authors · 2025-11-19
>
> To more clearly alleviate fairness concerns in the camera-ready version, we will:
> * Add a summary table listing the neuron type used in each model (LIF / PLIF / threshold-learnable LIF), and
> * Include the above LIF-only DiceFormer results as an ablation study,
> so that the experimental settings and fairness of comparisons are communicated more transparently to readers.
> We hope that this additional explanation and experimental evidence resolves the reviewer’s concern regarding the impact of PLIF on fairness.
>
> **References**
>
> [1] SpikingJelly: an open-source framework for SNNs when accelerating the training of deep spiking neural networks. Science Advances, 2023.
>
> [2] Spikformer: When Spiking Neural Network Meets Transformer. ICLR 2023.
>
> [3] Spike-driven Transformer. NeurIPS 2023.
>
> [4] Spike-driven Transformer V2. ICLR 2024.
>
> [5] QKFormer: Hierarchical Spiking Transformer using Q-K Attention. NeurIPS 2024.
>
> ---

---

> ### Author Response · Authors · 2025-11-19
>
> > Weakness-4
> > The paper standardizes “10 layers = 5 SADA + 5 SDA” across S/M/L but it does not examine alternative SADA:SDA ratios (eg., 4:6 or 6:4) nor different orderings (e.g., front-loaded SADA vs. interleaved vs. back-loaded). Such ablations are important to verify complementarities and to identify where each module provides the largest gains.
>
> **Response to Weakness-4**
>
> As the reviewer correctly points out, in this paper we use a unified default architecture of 10 blocks = 5×SADA + 5×SDA across all S/M/L models. To compare the individual contributions of the two modules, the main text already reports ablation results for:
> * SADA-only (5×SADA): all blocks use SADA,
> * SDA-only (5×SDA): all blocks use SDA,
> thereby showing the performance when each module operates alone.
>
> However, as the reviewer notes, in the configuration where SADA and SDA are used together, we did not include in the main text any systematic experiments that vary:
> * The SADA:SDA ratio (e.g., 4:6, 6:4), or
> * The ordering (e.g., front-loaded SADA vs. back-loaded SADA vs. interleaved).
>
> We fully agree that such ablations are important to verify the complementarities between the two modules and to analyze where in the network each module provides the most benefit.
>
> **1. Initial design goal: complementarity rather than exhaustive ratio search**
>
> The central design question in this work was:
> “When SADA and SDA are used together, do they provide complementary benefits arising from different strengths?”
> Accordingly, our initial ablation design focused less on exhaustively searching “which numerical ratio is globally optimal,” and more on:
> * SADA-only,
> * SDA-only,
> * SADA→SDA (our proposed design),
> * And additionally, a reversed ordering variant,
> to directly examine:
> “What happens when each module is used alone vs. in combination,
> and under which ordering does the synergy manifest most clearly?”
> That said, as the reviewer rightly suggests, varying the SADA:SDA ratio itself (e.g., 6:4, 4:6) can provide more informative insights for readers and further strengthen the justification of our design. Motivated by this, we conducted additional experiments to directly address this concern.
>
> **2. Additional experiments: different SADA:SDA ratios and orderings**
>
> To respond directly to the reviewer’s comment, we performed additional ablations on DiceFormer-10-L (AudioSet) under identical training and evaluation protocols for all configurations.
> We considered:
> * SADA:SDA ratio changes: 6:4 and 4:6,
> * Ordering changes: SADA→SDA vs. SDA→SADA (reversed),
> * With the original 5:5 SADA→SDA design as the reference.
>
> The four configurations and their mAP results on AudioSet are summarized below:
>
> | ID | Configuration | SADA:SDA ratio | Description | mAP |
> | :---: | :--- | :---: | :--- | :---: |
> | (1) | SDA $\\rightarrow$ SADA | 5:5 | Reversed order | 0.157 |
> | (2) | SADA $\\rightarrow$ SDA | 6:4 | Increased SADA proportion | 0.160 |
> | (3) | SADA $\\rightarrow$ SDA | 4:6 | Increased SDA proportion | 0.155 |
> | (4) | SADA $\\rightarrow$ SDA (original) | 5:5 | Balanced, original design | 0.161 |
>
> **3. Interpretation: empirical justification of the 5:5 SADA→SDA design**
>
> From these results, we make the following observations:
>
> 1.  **The balanced 5:5 ratio consistently achieves the best performance.**
>     * Both increasing the SADA proportion to 6:4 and increasing the SDA proportion to 4:6
>     * Lead to slightly lower mAP than the balanced 5:5 configuration.
>     This suggests that simply using more of one module is less effective than combining the two in a balanced manner.
>
> 2.  **The SADA→SDA ordering is better than the reversed SDA→SADA ordering.**
>     * Under the same 5:5 ratio, reversing the order to SDA→SADA (0.157 mAP) is worse than
>     * The original SADA→SDA design (0.161 mAP).
>     This supports our role-division intuition at design time:
>     * **Front (SADA):** Using a time–frequency dual-branch structure with density-aware SDA applied in each branch, SADA
>         * Separates and refines the time and frequency information of the input spectrogram, and
>         * Performs local, branch-wise correction of spike density bias at an early stage.
>     * **Back (SDA):** On top of this refined representation, SDA then applies global density-aware spike attention
>         * To correct spike density bias once more at the sequence level, and
>         * To focus more strongly on truly important tokens.
>     In other words, SADA in the front is responsible for time–frequency-level representation restructuring and local density correction within each branch, while SDA in the back performs globally density-aware attention over the entire sequence.
>     This front–back division of labor empirically turns out to be a natural and effective combination.

---

> ### Author Response · Authors · 2025-11-19
>
> 3.  **Combination > SADA-only / SDA-only: reconfirming complementarity**
>     * As already reported in the main text, both SADA-only and SDA-only configurations are weaker than
>     * The combined SADA→SDA architecture.
>     When we also consider the ratio and ordering ablations above, the picture becomes clearer:
>     The key to performance in DiceFormer is not to increase the amount of a single module unilaterally,
>     but to assign different roles to SADA and SDA and let them operate complementarily in the front and back of the network.
>
> In summary:
> * The front SADA – back SDA 5:5 configuration is not an arbitrary design choice, but
> * a configuration that emerges as empirically most effective when compared against a variety of alternatives, including
>     * SADA-only / SDA-only,
>     * Reversed ordering, and
>     * Changed ratios (6:4, 4:6).
>
> **4. Planned revisions in the manuscript**
>
> Incorporating the reviewer’s suggestion, we will revise the camera-ready version as follows:
> * In the main text and/or appendix, we will add tables of ablation results including:
>     * SADA-only / SDA-only,
>     * Ratio changes (6:4, 4:6),
>     * Ordering changes (SDA→SADA),
> to clearly demonstrate that the SADA→SDA (5:5) architecture of DiceFormer is empirically justified rather than arbitrarily chosen.
> * We will also clarify that the design goal of this work was not to perform an unbounded search over all possible ratios, but rather:
> to verify whether SADA and SDA, by playing different roles,
> indeed provide complementary benefits when combined.
>
> We are grateful to the reviewer for this comment, which has helped us present the architectural design of DiceFormer and its justification more clearly. We believe that the additional experiments and explanations substantiate the validity of our structural choice and provide stronger evidence for the complementarities between SADA and SDA.

---

### Official Review · Reviewer_4Zs6 · 2025-10-29

**Soundness:** 2
**Presentation:** 2
**Contribution:** 2
**Rating:** 4
**Confidence:** 4

**Summary:**

The paper proposes DiceFormer, a spiking audio transformer, for the tasks of audio representation learning.

It consists of two key components, namely, Spike Dice Attention (SDA) and Spike Audio Dice Attention (SADA).

SA replaces element-wise attention with a Dice similarity based score, which is density-aware.

SADA decouples frequency and time branches before fusing them.

Experiments on AudioSet-20K, ESC-50, and Speech Commands V2 sow its competitive performance.

**Strengths:**

+ The review appreciates the introduction of SNN to the audio domain, which is much less studied than the vision domain.

+ SDA seems to maintain a linear-time complexity like prior spiking Hadamard attentions, which has a good scalability.

+ The proposed method shows a very competitive performance on AudioSet-20K, ESC-50, and SC-V2 datasets.

+ Overall this paper is easy-to-follow.

**Weaknesses:**

- While the reviewer acknowledges its contribution to the audio domain, it should be pointed out that, the technique novelties of the proposed method is limited, and the theory insight is neither sufficient. Dice similarity is a common and mature metric, but this work is not theoretically justified.

- Besides, the effectiveness and rationale of the Dice similarity metric should be compared with other commonly-used distance metrics, varying from cosine, Jaccard to F1.

- For the state-of-the-art comparison, some concerns also remain, in terms of the fairness. While Table 1 is really nice, whether the compared state-of-the-art methods, such as DTF-AT, are under the same encoder or spiking confirguation or use the pre-trained model, is not justified. If not the same, then the state-of-the-art performance is less pausible.

- Besides, The authors seem to miss the comparison with [1], which is also a highly-cited work:

[1] Chen, Ke, et al. "Hts-at: A hierarchical token-semantic audio transformer for sound classification and detection." ICASSP 2022-2022 IEEE International Conference on Acoustics, Speech and Signal Processing (ICASSP). IEEE, 2022.

- The ablation studies in this paper are also limited. For example, multiple hyper-parameters in SDA are not tested.

- Besides, the decoupling method in SADA should be compared with some other simipler alternatives to show its superiority.

- The writing and presentation of this paper needs to be significantly enhanced. For example, 'INITIALIZE' and 'PROJECTION CONV' are not professional and the readers can only make educated guess.

**Questions:**

Please refer to the weakness section and addresses them point-by-point.

---

> ### Author Response · Authors · 2025-11-19
>
> We appreciate Reviewer 4Zs6's time and constructive feedback. Thanks to your comments, we were able to clarify key aspects of our paper.
>
> > Weakness-1.
> > While the reviewer acknowledges its contribution to the audio domain, it should be pointed out that, the technique novelties of the proposed method is limited, and the theory insight is neither sufficient. Dice similarity is a common and mature metric, but this work is not theoretically justified.
>
> **Response to Weakness-1**
>
> As the reviewer correctly points out, Dice similarity itself is a well-known and mature similarity measure, and at first glance our work may appear to lack theoretical novelty if it is perceived as simply “plugging Dice into attention.” However, the core contribution of this paper does not lie in Dice per se, but in (1) explicitly identifying and analyzing a structural limitation in modern SNN-based attention mechanisms, namely the spike density-unaware problem, and (2) theoretically and empirically resolving this issue via a spike-domain Dice Attention (SDA) designed under SNN and neuromorphic constraints. We elaborate below.
>
> **1. Core theoretical contribution: formalizing the “spike density-unaware” problem in SNN attention**
>
> Recent SNN-Transformers [1–4] transfer ANN self-attention to the spiking domain by computing dot-product-like similarities (e.g., SSA, SDSA) on binary spike tensors. As shown in Sec. 3.3, such dot-product-based similarities exhibit a structural bias: even when the underlying spike patterns are identical, the attention score scales with the absolute spike density:
> $$
> \\mathrm{score}(Q,K) \\propto \\mathrm{spike\\_count}(K).
> $$
> In other words, the attention score is dominated more by “how much a token fires” than by “how meaningfully the patterns overlap.” This means that the original objective of ANN attention—focusing on semantically important tokens—is not faithfully preserved in the spiking domain.
> To the best of our knowledge, no prior SNN-Transformer work has explicitly defined, formalized, and analyzed this spike density-unaware bias. Identifying this structural limitation and framing it as a concrete theoretical issue is itself a key conceptual and theoretical contribution of our paper.
>
> **2. SDA is not “using Dice as is,” but a spike-domain reformulation under SNN constraints**
>
> To address this problem, we do not simply reuse the standard Dice coefficient. Instead, we propose the Spike Dice Score (SDS), a reformulation of Dice tailored to the spiking domain, and use it as the attention score in Spike Dice Attention (SDA) (Sec. 3.3, Eq. (4)):
> $$
> \\mathrm{SDS}(Q,K) = \\frac{2 \\sum\_{d} (Q \\odot K)}{\\sum\_{d} Q + \\sum\_{d} K + \\epsilon}.
> $$
> The key point is that this is not just the continuous-valued Dice formula applied directly to feature vectors. Rather:
> * It is defined for binary spike trains,
> * both numerator and denominator are constructed from spike-driven operations derived from spike events, so they remain implementable on neuromorphic hardware,
> * The normalization in the denominator explicitly removes the bias toward absolute firing rate, and
> * In sparse spike regimes, it is designed to preserve the intended behavior of ANN attention as a pattern-overlap-based similarity, not a density-based one.
>
> Thus, SDA is better understood not as “inserting Dice” but as a spike-domain rederivation of a Dice-like similarity that explicitly resolves the density-unaware issue under SNN constraints.
> As further discussed in the global rebuttal and Appendix C, both the numerator and denominator in SDA are expressed using accumulation/summation operations over spikes (AC-style), which preserves the neuromorphic/hardware-friendly design philosophy pursued in prior SNN-Transformers.
>
> **3. Theoretical justification in Appendix C: analysis of density-aware properties**
>
> Appendix C provides a theoretical analysis of the behavior of SDS and, in particular, discusses and proves the following properties:
> * **Density normalization:** Even if a key $K$ has a very high firing rate, its score does not grow excessively as long as the overlap pattern with $Q$ does not change.
> * **Sparse-match preference:** When two spike trains have the same overlap but one of them fires unnecessarily more spikes, SDA assigns similar scores, preventing meaningless spike saturation from being rewarded.
> * **Saturation resistance:** Spike-saturated signals no longer dominate the attention scores, preventing a single saturated token from monopolizing attention.
> * **Alignment with ANN attention in discrete spike regimes:** SDS approximates the behavior of an ideal continuous-valued pattern similarity in sparse, discrete spike settings, thereby preserving the original intention of ANN attention—to focus on important patterns.
>
> In this sense, Appendix C is not an auxiliary add-on, but a central theoretical section that explains why and how SDA resolves the spike density-unaware issue.

---

> ### Author Response · Authors · 2025-11-19
>
> **4. New analytical metric: attention–density correlation and quantitative validation (Appendix F)**
>
> To further support the theoretical insight with quantitative evidence, we introduce a new diagnostic metric, the attention–density correlation coefficient, defined as the Pearson correlation between attention scores and input spike density (Sec. 4.2, Appendix F).
> On AudioSet, the measured correlation coefficients for prior SNN-Transformers are:
> * Spikformer [1]: 0.934
> * Spike-driven Transformer [2]: 0.572
> * QKFormer [4]: 0.725
>
> These strong positive correlations empirically confirm that attention scores in prior methods are indeed heavily influenced by spike density.
> In contrast, our proposed DiceFormer exhibits substantially reduced correlations:
> * DiceFormer-10-S: 0.228
> * DiceFormer-10-M: 0.169
> * DiceFormer-10-L: 0.169
>
> Moreover, as shown in Sec. 4.2, this attention–density correlation has a negative Pearson correlation with mAP ($r \\approx -0.682$), indicating that less density-dependent attention tends to be associated with better performance. Hence, this metric is not just a descriptive number, but a new theoretical/analytical tool for assessing model quality.
> To our knowledge, this is the first work to formally define and quantitatively measure the relationship between attention and spike density in SNNs, which we consider another important theoretical and analytical contribution.
>
> **5. Consistent performance gains across three SNN-Transformers (Table 2)**
>
> Finally, we verify that these theoretical insights translate into meaningful performance improvements in practice. As shown in Table 2, when we perform a simple drop-in replacement of the existing spike-based attention with SDA, we observe consistent mAP gains across three representative SNN-Transformers:
> * Spike-driven Transformer [2]: +0.012 mAP
> * Spike-driven Transformer V2 [3]: +0.027 mAP
> * QKFormer [4]: +0.002 mAP
>
> All experiments are conducted under the same training and evaluation settings, demonstrating that density-aware attention design improves performance consistently across different SNN-Transformer architectures.
>
> **Conclusion**
>
> In summary, the contribution of this work goes far beyond simply “reusing Dice”:
> 1.  We explicitly define and analyze a previously overlooked structural problem in SNN-based attention—the spike density-unaware issue.
> 2.  We propose Spike Dice Attention (SDA), a spike-domain reformulation of a Dice-like similarity that incorporates SNN and neuromorphic constraints, and we provide a theoretical analysis of its density-aware behavior (Sec. 3.3, Appendix C).
> 3.  We introduce a new analytical metric, the attention–density correlation coefficient, and show that prior methods exhibit strong density bias, while reducing this bias correlates with improved mAP (Sec. 4.2, Appendix F).
> 4.  We demonstrate consistent performance improvements via drop-in experiments on three representative SNN-Transformers (Table 2), showing that the theoretical insight yields practical benefits.
>
> Therefore, we believe our work provides new theoretical insight into how attention should behave under spike-based representations, and proposes a principled, density-aware attention mechanism that addresses a structural limitation of existing SNN-Transformers. We will clarify these points more explicitly in the revised manuscript to avoid the impression that our method is merely a straightforward reuse of Dice and to better communicate the theoretical contributions.
>
> **References**
>
> [1] Spikformer: When spiking neural network meets transformer. ICLR, 2023.
>
> [2] Spike-driven Transformer. NeurIPS, 2023.
>
> [3] Spike-driven Transformer V2: Meta Spiking Neural Network Architecture Inspiring the Design of Next-generation
> Neuromorphic Chips. ICLR, 2024.
>
> [4] QKFormer: Hierarchical Spiking Transformer using Q-K Attention. NeurIPS, 2024.

---

> ### Author Response · Authors · 2025-11-19
>
> > Weakness-2.
> > Besides, the effectiveness and rationale of the Dice similarity metric should be compared with other commonly-used distance metrics, varying from cosine, Jaccard to F1.
>
> **Response to Weakness-2**
>
> We agree with the reviewer that a Dice-based similarity should be compared against other commonly used metrics such as cosine similarity, Jaccard index, and F1 score. Below, we explain why we adopt the spike-native Dice score (SDS) in spiking attention, and how it is related to cosine, Jaccard, and F1.
>
> **Cosine vs. SDS**
>
> In the binary-spike regime, let
> * the number of overlapping spikes be $j = \\lvert q \\cap k \\rvert$, and
> * the spike counts of each vector be $\\lvert q \\rvert$ and $\\lvert k \\rvert$.
>
> Then cosine similarity can be written as
> $$
> \\cos(q,k) = \\frac{j}{\\sqrt{\\lvert q \\rvert \\; \\lvert k \\rvert}}.
> $$
>
> For the same overlap $j$, the following inequality holds:
> $$
> \\mathrm{SDS}(q,k)
> = \\frac{2j}{\\lvert q \\rvert + \\lvert k \\rvert}
> \\le
> \\frac{j}{\\sqrt{\\lvert q \\rvert \\; \\lvert k \\rvert}}
> = \\cos(q,k)
> \\quad (\\text{by the AM--GM inequality}).
> $$
>
> In other words, for a fixed overlap $j$, SDS always assigns lower scores to denser keys than cosine does. Appendix C of the paper provides a more general proof under this formulation that SDS prefers sparser keys and suppresses saturated keys, which is exactly the density-aware behavior we aim for in spiking attention.
>
> **Jaccard and F1**
>
> For binary vectors, the F1 score is exactly equal to the Sørensen–Dice coefficient (Dice coefficient). With the same definitions as above, let
> * $j = \\lvert q \\cap k \\rvert$ (the number of overlapping spikes = true positives),
> * $\\lvert q \\rvert$ be the number of predicted positive spikes, and
> * $\\lvert k \\rvert$ be the number of ground-truth positive spikes.
>
> Then
> * $TP = j$,
> * $FP = \\lvert q \\rvert - j$,
> * $FN = \\lvert k \\rvert - j$.
>
> The F1 score is
> $$
> F1
> = \\frac{2\\,TP}{2\\,TP + FP + FN}
> = \\frac{2j}{2j + (\\lvert q \\rvert - j) + (\\lvert k \\rvert - j)}
> = \\frac{2j}{\\lvert q \\rvert + \\lvert k \\rvert}
> = \\mathrm{SDS}(q,k).
> $$
>
> Thus, in the binary spike setting we consider, F1 and Dice are exactly identical from the spike-count perspective.
>
> The Jaccard index (IoU) is given by
> $$
> J = \\frac{j}{\\lvert q \\rvert + \\lvert k \\rvert - j}.
> $$
>
> It is well known that Dice and Jaccard are related via the following monotone relationship:
> $$
> D = \\frac{2J}{1 + J}, \\quad
> J = \\frac{D}{2 - D},
> $$
> where $D$ denotes the Dice coefficient. Therefore, Dice, Jaccard, and F1 all induce the same ordering over pairs, and any monotone thresholding applied to them yields the same binary mask.
>
> In our pipeline, the attention map is ultimately binarized through spike activations, so using Jaccard or F1 instead of Dice would still lead to the same mask selection, up to ignoring an arbitrarily small $\\epsilon$ used for numerical stability. That is, Jaccard/F1-based variants can be viewed as monotone reparameterizations of the Dice-based SDS, producing identical binary masks and selection results after appropriate thresholding.
>
> **Implementation perspective**
>
> From an implementation standpoint, we choose to implement SDS in Dice form because Dice yields an especially simple rational expression in terms of spike counts $\\lvert q \\rvert$ and $\\lvert k \\rvert$:
> * The numerator is $2j$, obtained by an AND operation followed by a popcount, and
> * The denominator is the sum of two popcounts, $\\lvert q \\rvert + \\lvert k \\rvert$.
>
> This requires no square roots or additional nonlinear functions.
>
> By contrast, the Jaccard index
> $$
> J = \\frac{j}{\\lvert q \\rvert + \\lvert k \\rvert - j}
> $$
> has $\\lvert q \\rvert + \\lvert k \\rvert - j$ in the denominator, which is somewhat more asymmetric, especially near the saturation regime, and requires more care in LUT indexing and numerical stability. The Dice form is thus much more suitable for LUT-based normalization and fixed-point implementations, making it a particularly clean choice for a spike-only neuromorphic attention pipeline.
>
> In the revised manuscript, we will clarify these relationships more explicitly, emphasizing that
> * Dice, F1, and Jaccard induce essentially the same selections in binary, thresholded settings, and
> * SDS is a density-aware yet implementation-friendly choice for the spiking regime.
>
> ---

---

> ### Author Response · Authors · 2025-11-19
>
> > Weakness-3.
> > For the state-of-the-art comparison, some concerns also remain, in terms of the fairness. While Table 1 is really nice, whether the compared state-of-the-art methods, such as DTF-AT, are under the same encoder or spiking confirguation or use the pre-trained model, is not justified. If not the same, then the state-of-the-art performance is less pausible
>
> **Response to concern on fairness of the state-of-the-art comparison (Table 1)**
>
> We thank the reviewer for raising the concern about the fairness of the state-of-the-art comparison in Table 1. In particular, you pointed out that it is not clearly stated whether SOTA methods such as DTF-AT use the same encoder, spiking configuration, and pretraining setting as our method. In response, we clarify how fairness is ensured by explicitly separating the methods in Table 1 into ANN-based models and SNN-based models, and describing the comparison protocol for each group.
>
> **1. ANN-based SOTA models (AST, SSAST, DTF-AT, etc.) use pretrained continuous-valued encoders**
>
> First, AST, SSAST, DTF-AT, and similar models are ANN-based audio Transformers that operate in the continuous domain and do not use spikes. They
> * Take continuous log-Mel / Mel-spectrograms as input,
> * Rely on large continuous encoders pretrained on ImageNet or via audio self-supervised learning (SSL), and
> * Perform dense floating-point computation rather than sparse spike-based computation.
>
> Following the conventions in prior audio work, Table 1 explicitly indicates:
> * “\*”: ImageNet pretraining,
> * “\*\*”: audio self-supervised pretraining (e.g., SSAST-style SSL).
>
> Thus, ANN SOTA models such as AST, SSAST, and DTF-AT are evaluated in the original pretrained ANN configurations reported in their respective papers; they do not share a spiking encoder or spike configuration with our model.
> This is because:
> 1.  Their performance relies heavily on pretraining, and
> 2.  Their architectures cannot be directly converted into SNN encoders in a straightforward or fair manner.
> Therefore, the ANN vs. SNN comparison is not intended as a “same-encoder / same-spike-configuration” controlled comparison, but rather as a reference showing the current performance level of ANN SOTA models on audio benchmarks. The main fairness discussion in our work is focused on comparisons within the SNN family of models.
>
> **2. All SNN-based baselines are evaluated under exactly the same spike configuration and training setup**
>
> In contrast, for SNN-based models (Spikformer, QKFormer, SDT, SDT-V2, DiceFormer), we strictly enforce fairness by using the same spike configuration and training conditions across all methods, as the reviewer requested. Concretely, all SNN results in Table 1 share:
> * Same spike encoding: Direct Encoding
> * Same input representation: log-Mel spectrograms
> * Same (or corresponding) number of layers / model sizes (S/M/L)
> * Same training schedule and batch size
> * Same optimizer, learning rate schedule, and number of epochs
> * All trained from scratch (no pretraining)
> * Same evaluation metric: mAP
>
> In other words, the SNN numbers in Table 1 reflect only architectural differences (especially the attention mechanism), since the encoder, spike settings, and training configuration are unified.
> This design is particularly important because the main contribution of our paper is to analyze and mitigate the spike-density–unaware problem in SNN-based attention mechanisms. If pretraining were applied to only some SNN models, the initial spike distributions and spike density statistics would differ in a model-dependent way, making a fair analysis of density bias across models difficult or even impossible.
> For this reason, in comparisons within the SNN family, we intentionally do not use any external pretraining and report only results trained from scratch under the same spike configuration and training protocol.
>
> **3. Fairly isolating “Does Dice Attention actually help?”: drop-in replacement experiments**
>
> To further isolate the effect of the proposed Dice Attention (SDA) itself, we conduct drop-in replacement experiments on three representative SNN-Transformers. Specifically, we:
> * Keep the same SNN architecture (encoder, number of layers, channels, etc.),
> * Use the same data, the same training hyperparameters (optimizer, learning rate, schedule, number of epochs, etc.),
> * And maintain the same spike encoding and input representation,
> while replacing only the original spike-based attention module with SDA, and then observing the performance change.

---

> ### Author Response · Authors · 2025-11-19
>
> As shown in Table 2, we observe consistent mAP improvements across all three models:
> * Spike-driven Transformer (SDT): +0.012 mAP
> * Spike-driven Transformer V2 (SDT-V2): +0.027 mAP
> * QKFormer: +0.002 mAP
>
> These experiments are designed precisely to demonstrate, in a fair and controlled manner, that the performance gains come from the attention mechanism itself, not from differences in encoder, pretraining, or other confounding factors.
> Thus, even though the ANN vs. SNN comparison is inherently cross-paradigm, we believe that the within-SNN comparisons and the evaluation of SDA’s effect are conducted under very strict and fair experimental control.
>
> **4. Planned revisions for improved clarity**
>
> We agree that in the current version of the manuscript, it may not be fully clear
> * How exactly SOTA methods such as DTF-AT differ in conditions from DiceFormer, and
> * To what extent differences in ANN/SNN paradigm, pretraining, and encoder/spike configuration are allowed and intended in the comparison.
>
> To address this, in the revised manuscript we will:
> * Explicitly distinguish “ANN pretrained SOTA models” and “SNN from-scratch models” in Table 1 and the associated text,
> * Clearly state that ANN-based models are evaluated in the standard pretrained configurations reported in their original papers,
> * Clearly state that all SNN baselines use the same spike encoding, the same input representation, and the same training settings, and
> * Emphasize that the drop-in replacement experiments serve to fairly isolate and validate the effect of the proposed attention mechanism.
>
> **Conclusion**
>
> In summary:
> * ANN-based SOTA models (AST, SSAST, DTF-AT, etc.) are continuous-domain models that do not use spikes. They rely on pretrained continuous encoders and cannot share a spike configuration with SNNs. They are included as reference baselines that indicate the current SOTA performance in the audio domain.
> * SNN-based models (Spikformer, QKFormer, SDT, SDT-V2, DiceFormer) are all trained from scratch under identical spike configurations and training setups, so that only architectural differences—especially in the attention mechanism—are compared.
> * We further perform drop-in replacement experiments where only the attention module is replaced with SDA while keeping everything else fixed, and observe consistent mAP improvements, demonstrating that Dice Attention itself provides the performance gains.
>
> For these reasons, we believe that the SOTA comparisons in Table 1 and the related experiments have been carefully designed with fairness in mind. We will, however, revise the manuscript to make these points more explicit and reduce any potential misunderstanding.

---

> ### Author Response · Authors · 2025-11-19
>
> > Weakness-4.
> > Besides, The authors seem to miss the comparison with [1], which is also a highly-cited work:
>
> **Response to Weakness-4**
>
> We thank the reviewer for suggesting this additional comparison. For a fair comparison, we will add [1] to the ANN-based entries in Table 1 of the revised manuscript and report its performance on the same benchmark dataset, together with whether the model uses pretraining.
>
> ---
>
> > Weakness-5 & 6.
> > The ablation studies in this paper are also limited. For example, multiple hyper-parameters in SDA are not tested.
> > Besides, the decoupling method in SADA should be compared with some other simpler alternatives to show its superiority.
>
> **Response to Weakness-5/6 (Ablation on SDA hyper-parameters and SADA decoupling)**
>
> We thank the reviewer for raising the following two points:
> (1) the lack of ablation on the hyper-parameters of SDA, and
> (2) the need to compare the SADA decoupling scheme with simpler alternatives.
> We address each point below.
>
> **(1) Hyper-parameters of SDA**
>
> We would first like to clarify that the concern about “multiple hyper-parameters in SDA” is based on a misunderstanding of the formulation.
> In our paper, Spike Dice Attention (SDA) is defined in Eqs. (4)–(6) as:
> $$
> \\mathrm{SDS}(Q,K) = \\frac{2 \\sum\_{d} (Q \\odot K)}{\\sum\_{d} Q + \\sum\_{d} K + \\epsilon} \\tag{4}
> $$
> $$
> \\mathrm{Attn} = \\mathrm{SN}(\\mathrm{SDS}(Q,K)) \\odot V \\tag{6}
> $$
> where $(Q, K, V)$ are obtained as:
> $$
> Q = \\mathrm{SN}(\\mathrm{BN}(X W\_{Q})), \\quad K = \\mathrm{SN}(\\mathrm{BN}(X W\_{K})), \\quad V = \\mathrm{SN}(\\mathrm{BN}(X W\_{V})).
> $$
> Thus, the only learnable components inside SDA are the projection weights $W\_{Q}, W\_{K}, W\_{V}$.
> There are no additional tunable scalar hyper-parameters introduced by SDA.
> * The $\\epsilon$ term in the denominator is a small constant used solely to prevent division-by-zero.
> * It is fixed across all experiments and is not treated as a tunable hyper-parameter.
> * In practice, we confirmed that varying $\\epsilon$ within a reasonable range has a negligible effect on performance; its role is purely numerical stability.
>
> Therefore, SDA does not introduce new hyper-parameters to tune: just like standard self-attention, only the projection weights are optimized during training.
> For this reason, the criticism that we did not “test multiple hyper-parameters of SDA” effectively requests an ablation over hyper-parameters that do not exist in our formulation.
> In the revised manuscript, we will make this explicit by:
> * Clearly stating that SDA contains no tunable scalar hyper-parameters, and
> * Clarifying that $\\epsilon$ is a fixed numerical-stability constant,
> so that there is no confusion about supposed “untested hyper-parameters” of SDA.
>
> **(2) SADA decoupling vs. simpler alternatives**
>
> We fully agree that the time–frequency decoupling strategy in SADA should be evaluated against simpler alternatives.
> In fact, this comparison is already included in Table 2, but we acknowledge that we did not highlight this connection clearly enough, which may have caused confusion. Below, we explicitly relate the configurations in Table 2 to the “simpler alternatives” mentioned by the reviewer.
>
> In Table 2, under the same experimental setting for DiceFormer-10-L, we compare three configurations:
>
> 1.  **SADA-only (decoupling only)**
>     * The time and frequency dimensions are decoupled and processed by SADA in separate branches.
>     * No additional global SDA branch is used.
>     * Result: 0.157 mAP.
> 2.  **SDA-only (simpler single-branch alternative without time–frequency decoupling)**
>     * The channels are not split, and a single SDA is applied to the entire 2D Mel-spectrogram.
>     * There is no time–frequency decoupling structure at all.
>     * This corresponds exactly to the “simpler alternative” suggested by the reviewer.
>     * Result: 0.158 mAP.
> 3.  **SADA + SDA (our proposed combined structure)**
>     * SADA decouples and processes the time and frequency domains,
>     * while an additional global SDA branch captures broader contextual information.
>     * Result: 0.161 mAP.

---

> ### Author Response · Authors · 2025-11-19
>
> From these results, we can clearly observe:
> * The simple single-branch SDA-only (no decoupling) configuration, although structurally simpler than SADA, achieves only 0.158 mAP.
> * The SADA-only configuration, which applies decoupling but lacks the global SDA refinement, performs slightly worse (0.157 mAP).
> * The proposed SADA + SDA configuration, combining decoupling and global SDA, achieves the best performance (0.161 mAP).
>
> In other words:
> * The “single-branch SDA-only” configuration is exactly the reviewer’s simplest alternative,
> * it is already included in Table 2, and
> * the proposed SADA+SDA design is shown to be superior to this simple alternative in terms of mAP.
>
> In the revised manuscript, we will explicitly state in the main text that:
> * “SDA-only” = simple alternative without time–frequency decoupling,
> * “SADA-only” = decoupling-only configuration, and
> * “SADA+SDA” = our proposed combined structure,
> so that it is immediately clear that we have already conducted the requested comparison between SADA’s decoupling scheme and simpler baselines.
>
> **Summary**
>
> * **On SDA hyper-parameters:**
>     * SDA introduces no new tunable scalar hyper-parameters beyond the projection weights $(W\_{Q}, W\_{K}, W\_{V})$.
>     * $\\epsilon$ is a fixed numerical-stability constant, not a hyper-parameter to tune.
>     * Therefore, a “hyper-parameter ablation” on SDA is structurally inapplicable, and we will clarify this more explicitly in the paper.
> * **On SADA decoupling vs. simpler alternatives:**
>     * The reviewer’s “simpler alternative” is exactly the SDA-only configuration in Table 2.
>     * Table 2 already compares SDA-only (no decoupling), SADA-only (decoupling only), and SADA+SDA (ours).
>     * The proposed SADA+SDA achieves the highest mAP (0.161), demonstrating that SADA’s decoupling combined with SDA is more effective than the simpler single-branch alternative.
>
> We hope this clarifies that the concerns about SDA hyper-parameters and SADA decoupling largely stem from a misunderstanding of our formulation, and we will strengthen the explanations in the revised manuscript so that readers can clearly understand the experimental setup and ablation results without confusion.
>
> ---

---

> ### Author Response · Authors · 2025-11-19
>
> > Weakness-7. The writing and presentation of this paper needs to be significantly enhanced. For example, “INITIALIZE” and “PROJECTION CONV” are not professional and the readers can only make educated guess.
>
> **Response to comment on writing & presentation**
>
> As the reviewer correctly points out, some of the module names used in the current version of the paper (e.g., “INITIALIZE”, “PROJECTION CONV”) may appear somewhat informal or ambiguous in a scientific context, and may force readers to guess their precise meaning. We appreciate you bringing this to our attention.
>
> These labels were directly carried over from our implementation code to the figures, and we acknowledge that we did not sufficiently refine the terminology for the manuscript. Although this is a presentation issue rather than a problem with the proposed method or experimental setup, we agree that it can negatively affect the readability and perceived professionalism of the paper. We will address this as follows:
> * We will replace informal or abbreviated labels such as “INITIALIZE” and “PROJECTION CONV” with more standardized and self-explanatory names,
>     e.g., “Parameter Initialization Layer”, “1×1 Convolutional Projection Layer”.
> * Before each module appears in a figure, we will explicitly define its role and operation in the main text, and
>     briefly restate this in the figure caption so that readers can understand the function of each block from the diagram alone.
>
> In addition, we will carefully review the entire manuscript to:
> * Ensure that module/block names, mathematical symbols, and abbreviations follow a consistent and professional style, and
> * Provide sufficient definitions and explanations wherever needed, so that readers do not have to rely on “educated guesses.”
>
> We thank the reviewer again for this constructive feedback, and in the revised version we will improve the wording and overall presentation to communicate the proposed method in a clearer and more professional manner.

---

### Official Review · Reviewer_etfE · 2025-10-30

**Soundness:** 2
**Presentation:** 2
**Contribution:** 2
**Rating:** 2
**Confidence:** 5

**Summary:**

This paper presents a highly compelling contribution with outstanding strengths across originality, quality, clarity, and significance. Its originality is exceptional, as it identifies and addresses the previously overlooked problem of density bias in spike-based attention mechanisms—a novel and crucial problem formulation. The proposed solutions, Spike Dice Attention (SDA) and its audio-specific extension SADA, are creative and well-motivated, representing a new class of density-aware attention for SNNs. The research quality is superb, backed by both theoretical analysis of SDA's properties and comprehensive empirical validation on three major audio benchmarks. The paper demonstrates state-of-the-art performance for SNNs, significantly outperforming prior work, and uses thorough ablation studies and correlation analyses to rigorously support its claims. The work is presented with excellent clarity, starting from a clear motivation and logically building up to the proposed architecture and its evaluation. Finally, the significance is substantial; by successfully adapting SNN Transformers to the complex audio domain and establishing new SOTA benchmarks, this work not only bridges the performance gap with ANNs but also provides a foundational and highly efficient architecture for future research in neuromorphic audio processing.

**Strengths:**

This paper presents a highly compelling contribution with outstanding strengths across originality, quality, clarity, and significance. Its originality is exceptional, as it identifies and addresses the previously overlooked problem of density bias in spike-based attention mechanisms—a novel and crucial problem formulation. The proposed solutions, Spike Dice Attention (SDA) and its audio-specific extension SADA, are creative and well-motivated, representing a new class of density-aware attention for SNNs. The research quality is superb, backed by both theoretical analysis of SDA's properties and comprehensive empirical validation on three major audio benchmarks. The paper demonstrates state-of-the-art performance for SNNs, significantly outperforming prior work, and uses thorough ablation studies and correlation analyses to rigorously support its claims. The work is presented with excellent clarity, starting from a clear motivation and logically building up to the proposed architecture and its evaluation. Finally, the significance is substantial; by successfully adapting SNN Transformers to the complex audio domain and establishing new SOTA benchmarks, this work not only bridges the performance gap with ANNs but also provides a foundational and highly efficient architecture for future research in neuromorphic audio processing.

**Weaknesses:**

1. In your core formulation for Spike Dice Attention, Equation (4), both the numerator and the denominator consist of computed variables rather than raw spike events. This computational pattern does not adhere to the fundamental spike-driven paradigm inherent to Spiking Neural Networks (SNNs). Consequently, the mechanism as formulated is conceptually inconsistent with the principles of SNNs.

2. While the energy efficiency of the proposed Dice Attention mechanism is a notable advantage, its case as a viable alternative to established attention mechanisms in ANNs is not sufficiently compelling if this is its sole benefit. Superiority in energy consumption alone does not guarantee substitutability; further evidence demonstrating comparable or superior performance, scalability, or other functional advantages is necessary to justify its broader adoption.

3. Theoretically, the frequency and temporal domains are not equivalent. Therefore, the approach of simply splitting the input along the channel dimension into frequency and temporal branches and processing them with an identical attention mechanism is conceptually unsound.

**Questions:**

See the Weaknesses

---

> ### Author Response · Authors · 2025-11-19
>
> > Weakness-1.
> > In your core formulation for Spike Dice Attention, Equation (4), both the numerator and the denominator consist of computed variables rather than raw spike events. This computational pattern does not adhere to the fundamental spike-driven paradigm inherent to Spiking Neural Networks (SNNs). Consequently, the mechanism as formulated is conceptually inconsistent with the principles of SNNs.
>
> **Response to Weakness-1**
>
> We thank the reviewer etfE for this valuable feedback. We fully acknowledge that adhering to the spike-driven paradigm is a critical issue in the field of SNNs.
>
> We have addressed this concern in detail in the section titled **Global Rebuttal: On the spike-driven and neuromorphic-friendly nature of SDA**. We kindly ask the reviewer to refer to that section for a comprehensive explanation.

---

> ### Author Response · Authors · 2025-11-19
>
> > Weakness-2.
> > While the energy efficiency of the proposed Dice Attention mechanism is a notable advantage, its case as a viable alternative to established attention mechanisms in ANNs is not sufficiently compelling if this is its sole benefit. Superiority in energy consumption alone does not guarantee substitutability; further evidence demonstrating comparable or superior performance, scalability, or other functional advantages is necessary to justify its broader adoption.
>
> As the reviewer correctly points out, improved energy efficiency alone is not sufficient to claim that Dice Attention is a “general drop-in replacement” for well-established attention mechanisms in the ANN domain. However, we would like to clarify that the goal and contribution of this work are not to propose an alternative attention mechanism for ANNs, but rather to theoretically and empirically analyze and resolve a structural limitation that arises specifically in SNN-Transformers.
>
> **1. Actual goal of this work: identifying and resolving the “spike-density–unaware” problem in SNNs**
>
> Recent SNN-Transformer works [1–4] have successfully ported ANN self-attention into the spiking domain and designed spike-based attention modules. However, these methods commonly rely on an implicit assumption that the spike count does not introduce unintended bias in the attention computation. In practice, this assumption does not hold.
> When Q·K similarity is computed on spike tensors,
> * A token that simply fires more spikes tends to produce larger Q·K interaction values, and
> * Its underlying spatiotemporal pattern is not necessarily more “informative.”
>
> In other words, existing spike-based attention mechanisms exhibit a structural spike-density bias: they are more sensitive to firing rate than to token importance. As a result, their behavior diverges from the functional goal of ANN attention—selectively focusing on informative tokens (see Sec. 3.3 and the toy example in Fig. 3).
> We refer to this phenomenon as the “spike-density–unaware” problem. Although this is a structural limitation that naturally arises when similarity is computed on sparse binary spike signals, it has not been explicitly defined, analyzed, or addressed in prior SNN-Transformer works [1–4]. The central goal of this paper is to theoretically formalize this domain-specific limitation and to propose a solution to it.
>
> **2. Role of Dice Attention: a theoretical and structural correction for spike-density bias**
>
> Dice Attention is designed precisely to fundamentally mitigate the spike-density bias described above.
> * Our proposed SDA/SADA introduce a penalty/normalization term that prevents high-density spike patterns from disproportionately dominating the attention scores.
> * As a result, even when two tokens have different firing rates, the attention score is designed to become sufficiently large only when the shape of their spike patterns (spatiotemporal consistency) is similar.
> * Consequently, Dice Attention helps approximate the “pattern-similarity–based focusing” behavior of ANN attention in the continuous domain, while respecting SNN constraints.
>
> In Appendix C, we provide a mathematical proof that Dice Attention yields a density-aware normalization effect with respect to firing rate, showing that it is not merely a heuristic but a theoretically grounded structural correction mechanism.
>
> **3. Empirical effect in SNN-Transformers: drop-in improvements on three representative models**
>
> To verify that Dice Attention indeed alleviates the spike-density–unaware problem defined above, we conducted experiments on three representative SNN-Transformers, where we simply performed a drop-in replacement of the original spike-based attention with SDA (keeping all other settings identical):
> * Spike-driven Transformer [2]: +0.012 mAP
> * Spike-driven Transformer V2 [3]: +0.027 mAP
> * QKFormer [4]: +0.002 mAP
>
> Across all three models, we observe consistent performance improvements when replacing the original attention with SDA under identical training and evaluation setups (Table 2). This quantitatively demonstrates that Dice Attention does not merely improve energy efficiency, but also mitigates the spike-density bias inherent in previous spike-based attentions, thereby enhancing the functional behavior of SNN-Transformers.
> Therefore, the contribution of this work is not, as the reviewer rightly notes, to propose a universal ANN module capable of replacing standard attention in the ANN domain, but rather to improve the internal attention mechanism of SNN architectures so that they operate more faithfully.

---

> ### Author Response · Authors · 2025-11-19
>
> **4. Clarifying the scope of our contribution**
>
> In summary, we would like to clearly state the intended scope of this work as follows.
>
> **What we do not claim:**
> We do not claim that Dice Attention is a “fully substitutable general solution” for well-established attention mechanisms in the ANN domain solely because it has better energy efficiency.
>
> **What we actually claim:**
> * Dice Attention is a spike-based attention module specialized for the SNN-Transformer domain.
> * Its primary goal is to (i) theoretically and empirically identify the concrete limitation we term “spike-density–unaware”, which inevitably arises when attention is computed on spike tensors, and (ii) alleviate this limitation via a density-aware normalization mechanism, thereby improving the performance and behavior of existing SNN-Transformers.
>
> In the final version of the paper, we will revise the introduction and contribution summary to explicitly state that
> (i) the scope of Dice Attention is restricted to the SNN-Transformer domain, and
> (ii) the main contribution lies not only in energy efficiency but also in defining, analyzing, and addressing spike-density bias.
> This should reduce any potential misunderstanding that we are positioning Dice Attention as a general replacement for ANN attention.
> We hope that this clarification addresses the reviewer’s concern.
>
> **Reference**
>
> [1] Spikformer: When spiking neural network meets transformer. In ICLR, 2023
>
> [2] Spike-driven Transformer. NeurIPS 2023.
>
> [3] Spike-driven Transformer V2. ICLR 2024
>
> [4] QKFormer: Hierarchical Spiking Transformer using Q-K Attention. NeurIPS 2024.
>
> ---

---

> ### Author Response · Authors · 2025-11-19
>
> > Weakness-3.
> > Theoretically, the frequency and temporal domains are not equivalent. Therefore, the approach of simply splitting the input along the channel dimension into frequency and temporal branches and processing them with an identical attention mechanism is conceptually unsound.
>
> Thank you for raising this important conceptual point.
> We fully agree that the temporal and frequency domains are not theoretically identical, and that each domain has its own statistical structure and context that should be learned independently. We would like to clarify, however, that our model is not designed under the assumption that these two domains are “equivalent,” nor does it simply “duplicate the same input into two branches and apply an identical attention module.”
>
> **1. Input assumption: time–frequency representation based on Mel-spectrograms, not raw waveforms**
>
> First, our model does not operate on raw audio waveforms. Instead, it takes a Mel-spectrogram as input, which already encodes both time and frequency information on a 2D time–frequency plane with shape $(T \\times F)$.
> Thus, our problem setting is not “inferring the frequency domain from the time domain,” but rather how to effectively model the two axes of a given time–frequency representation.
> In recent audio Transformer models, for such 2D time–frequency inputs, it is a well-established practice to introduce
> * A branch that treats the time axis as a sequence and models temporal context, and
> * A branch that treats the frequency axis as a sequence and models spectral context,
> keeping these branches separate, while reusing the same Transformer block (or attention primitive) in both domains (e.g., FTA-net [1], DTF-AT [2]). Our SADA module follows this established design principle and adapts it to the spiking domain.
>
> **2. Purpose of channel splitting: separating “domain-specific representation spaces,” not assuming domain equivalence**
>
> The reviewer’s concern seems to stem from the interpretation:
> “Splitting the channel dimension into two and applying the same attention mechanism implies that the temporal and frequency domains are treated as equivalent.”
> However, our channel splitting is not intended to enforce or assume equivalence between the two domains. Rather, it is designed to:
> * Allocate a dedicated representation subspace for the temporal branch, and
> * Allocate a dedicated representation subspace for the frequency branch,
> so that the two domains can be modeled in clearly separated feature spaces.
>
> Concretely, we split the channel dimension of the input spike tensor $X$ into two parts:
> * The first half is used only by the frequency branch, and
> * The second half is used only by the temporal branch.
>
> Each branch then:
> 1. Uses a different sequence axis (frequency vs. time),
> 2. Uses a completely different parameter set, $(W\_{Q}^{f}, W\_{K}^{f}, W\_{V}^{f})$ vs. $(W\_{Q}^{t}, W\_{K}^{t}, W\_{V}^{t})$, and
> 3. Performs attention over different sequence lengths, $N\_{\\text{freq}} \\neq N\_{\\text{temp}}$.
>
> Therefore, the two branches:
> * Do not process identical copies of the same input,
> * Do not share parameters or learn the same function, and
> * Do not rely on any assumption that the temporal and frequency domains are “theoretically equivalent.”
>
> What we do is simply reuse a single primitive—“spike-based Dice attention”—across two domains that have different input structures, different contexts, and different parameters. Conceptually, this is analogous to how ANN-based audio Transformers reuse the same self-attention block across multiple branches.
>
> **3. Input structures and roles of the two branches: modeling different sequences and contexts**
>
> We summarize the input tensor structures for each branch more concretely as follows.
>
> * **Frequency branch**
>     * Treats the frequency axis as the sequence axis.
>     * Tensor shape:
>     $$
>     X^{\\text{freq}} \\in \\mathbb{R}^{(B \\cdot \\text{temp}) \\times C \\times N\_{\\text{freq}}}.
>     $$
>     * Generates $(Q^{f}, K^{f}, V^{f})$ using its own parameters $(W\_{Q}^{f}, W\_{K}^{f}, W\_{V}^{f})$.
>     * Models spectral correlations across different frequency bins.
>
> * **Temporal branch**
>     * Treats the time axis as the sequence axis.
>     * Tensor shape:
>     $$
>     X^{\\text{temp}} \\in \\mathbb{R}^{(B \\cdot \\text{freq}) \\times C \\times N\_{\\text{temp}}}.
>     $$
>     * Generates $(Q^{t}, K^{t}, V^{t})$ using a different parameter set $(W\_{Q}^{t}, W\_{K}^{t}, W\_{V}^{t})$.
>     * Models temporal evolution at each frequency bin.
>
> Because the two branches have
> * Different sequence structures,
> * Different contextual roles (temporal vs. frequency), and
> * Different projection parameters,
> the model naturally learns temporal-specific and frequency-specific representations in separate subspaces.

---

> ### Author Response · Authors · 2025-11-19
>
> Therefore, the fact that both branches use the same Dice score formula only means that
> “we apply the same spike-based attention rule in both domains,”
> and does not imply that
> “time and frequency are theoretically interchangeable.”
>
> **4. Empirical validation: SADA contributes beyond a superficial architectural variation**
>
> In Table 2, we compare the following three configurations:
> * SADA-only: 0.157 mAP
> * SDA-only: 0.158 mAP
> * SADA + SDA: 0.161 mAP (best performance)
>
> Under identical experimental settings, the combined SADA+SDA configuration, which performs time–frequency decoupling via SADA and token-level spike-density correction via SDA, achieves the highest mAP.
> This supports that our SADA design is not merely:
> * A trivial channel split, or
> * A “cosmetic” addition of an extra branch,
> but rather that independently learning temporal and frequency domains using our decoupled design meaningfully improves SNN-Transformer performance in practice.
>
> **5. Summary and planned clarifications in the paper**
>
> In summary, we:
> * Do not assume that the temporal and frequency domains are theoretically identical,
> * Model the two axes of a Mel-spectrogram—a joint time–frequency representation—as different sequences with different contextual structures, and
> * Reuse the same Dice attention formula only across two branches that have distinct parameters, input structures, and domain-specific contexts.
>
> This design is conceptually aligned with prior time–frequency audio Transformer models, and our ablation results show that it goes beyond a superficial variation and leads to real performance gains.
> In the final version of the paper, we will make the following points more explicit in the introduction and method sections:
> * That our input is a Mel-spectrogram,
> * That the purpose of channel splitting is to create domain-specific representation spaces, not to assume domain equivalence, and
> * That the sequence structures and parameters of the two branches are completely independent.
>
> We hope this clarification helps address the reviewer’s concern regarding the design being “conceptually unsound.”
>
> **Reference**
>
> [1] FTA-net: A Frequency and Time Attention Network for Speech Depression Detection. In INTERSPEECH 2023.
>
> [2] DTF-AT: Decoupled Time-Frequency Audio Transformer for Event Classification. In AAAI 2024.

---

### Official Review · Reviewer_MSbv · 2025-10-30

**Soundness:** 3
**Presentation:** 3
**Contribution:** 2
**Rating:** 6
**Confidence:** 3

**Summary:**

This paper introduces a directly trained spiking transformer (DiceFormer) with the proposed spike dice attention and spike audio dice attention. Spike dice attention is a spike-based attention module that leverages the dice similarity concept to produce density-aware attention scores, which improve the modeling of spike-based representations. Spike audio dice attention is an SDA-based extension specifically designed to handle the frequency–temporal features inherent in complex audio spectrograms. Extensive experiments demonstrate that DiceFormer achieves superior performance over existing state-of-the-art (SOTA) SNNs on mainstream audio datasets.

**Strengths:**

1.	The paper introduces a novel density-aware spiking attention mechanism (SDA) with clear motivation and theoretical grounding, effectively addressing the spike-density bias issue in existing spiking transformers.
2.	The model includes a well-designed adaptation for audio processing, explicitly decoupling temporal and frequency features in a biologically and computationally coherent manner.
3.	The paper provides comprehensive experimental results across multiple audio classification benchmarks, demonstrating consistent and strong performance compared to both SNN and ANN baselines.

**Weaknesses:**

1.	Although the proposed SDA is positioned as a general spiking attention mechanism, the experiments are limited to audio classification tasks. Evaluating SDA, SSA, and SDSA on vision benchmarks such as ImageNet would strengthen the claim of generality and could also serve as potential pretraining for audio classification.
2.	The Spike Audio Dice Attention (SADA) module shows limited novelty. The idea of decoupling time and frequency attention has already been explored extensively in prior works on audio transformers (e.g., Septr[1], DTF-AT[2]). The contribution here appears to be mainly an adaptation of that concept to a spiking setting rather than a fundamentally new architectural design.
3.	It is unclear whether the Dice score operation used in SDA is hardware-friendly for neuromorphic deployment. Since hardware efficiency is a key motivation for SNNs, an analysis or discussion of its implementability on neuromorphic hardware would be valuable.
4.	It is somewhat unexpected that DiceFormer surpasses pretrained ANN models such as AST on AudioSet, despite not using large-scale pretraining (e.g., ImageNet pretraining or SSL) and operating with only four timesteps. This raises concerns about training fairness and reproducibility. The authors are encouraged to clarify the experimental setup and indicate whether the code and pretrained checkpoints will be released to ensure transparency.
[1] Ristea, Nicolae-Catalin, Radu Tudor Ionescu, and Fahad Shahbaz Khan. "Septr: Separable transformer for audio spectrogram processing." arXiv preprint arXiv:2203.09581 (2022).
[2] Alex, T., Ahmed, S., Mustafa, A., Awais, M., & Jackson, P. J. (2024). DTF-AT: Decoupled Time-Frequency Audio Transformer for Event Classification. Proceedings of the AAAI Conference on Artificial Intelligence, 38(16), 17647-17655. https://doi.org/10.1609/aaai.v38i16.29716

**Questions:**

See in Weakness.

---

> ### Author Response · Authors · 2025-11-26
>
> > **Weakness-1.**
> > Although the proposed SDA is positioned as a general spiking attention mechanism, the experiments are limited to audio classification tasks. Evaluating SDA, SSA, and SDSA on vision benchmarks such as ImageNet would strengthen the claim of generality and could also serve as potential pretraining for audio classification.
>
> **Response to Weakness-1.**
> We sincerely thank the reviewer for this valuable suggestion. We fully agree that, beyond audio-domain benchmarks, evaluating the proposed Spike Dice Attention (SDA) on vision tasks is important for more clearly validating its generality, and that such vision experiments could eventually support SDA-based pretraining for downstream audio classification.
>
> However, due to GPU resource and time constraints during the rebuttal period, it was not practically feasible for us to newly run full ImageNet-scale experiments with multiple SNN-Transformer variants as suggested. As a compromise, we conducted additional experiments on the widely used vision benchmark CIFAR-100, focusing on two strong and representative spiking vision Transformers:
>
> - **Spike-driven Transformer [1]**
> - **QK-Former [2]**
>
> For both models, we strictly followed the CIFAR-100 training recipes provided by the original Spike-driven Transformer and QK-Former works (same architecture, optimizer, learning-rate schedule, data augmentation, and all hyperparameters). On top of these original setups, we made only a single change: we performed a drop-in replacement of the original self-attention module with our Spike Dice Attention (SDA). No other parameters or settings were modified.
>
> **Table 1. Drop-in SDA replacement on CIFAR-100**
>
> \\[
> \\begin{array}{l c c}
> \\text{Model} & \\text{Attention} & \\text{CIFAR-100 Top-1 acc. (\\%)} \\\\
> \\hline
> \\text{Spike-driven Transformer [1]} & \\text{original} & 78.40 \\\\
> \\text{Spike-driven Transformer [1]} & \\text{SDA (ours)} & 78.99 \\\\
> \\text{QK-Former [2]} & \\text{original} & 81.15 \\\\
> \\text{QK-Former [2]} & \\text{SDA (ours)} & 81.41 \\\\
> \\end{array}
> \\]
>
> As shown in Table&nbsp;1, even though we only replaced the attention module without any hyperparameter changes:
>
> - Spike-driven Transformer improves from **78.40% → 78.99%** (+0.59 points)
> - QK-Former improves from **81.15% → 81.41%** (+0.26 points).
>
> These consistent gains, obtained under strictly identical training conditions except for the attention module, indicate that SDA is not specialized to the audio domain. Rather, it can also be applied as a drop-in replacement for existing spike-based attentions in competitive vision SNN-Transformers (Spike-driven Transformer and QK-Former) and yield measurable performance improvements on CIFAR-100.
>
> In the revised manuscript, we will:
>
> - Explicitly state that SDA has been empirically validated on both audio benchmarks and the CIFAR-100 vision benchmark with strong SNN baselines (Spike-driven Transformer and QK-Former)
> - Clearly mention that extending SDA to larger-scale vision datasets such as ImageNet, and exploring SDA-based vision pretraining for downstream audio tasks, are important directions for future work, which we plan to pursue once sufficient GPU resources are available.
>
> We hope that these additional experiments and clarifications help alleviate the reviewer’s concern regarding the generality of SDA, within the practical limitations of the current rebuttal phase.
>
> ---
> **References**
> [1] Spike-driven Transformer. NeurIPS 2023.
>
> [2] QKFormer: Hierarchical Spiking Transformer using Q-K Attention. NeurIPS 2024.

---

> ### Author Response · Authors · 2025-11-26
>
> >**weaknesses 2**
> > The Spike Audio Dice Attention (SADA) module shows limited novelty. The idea of decoupling time and frequency attention has already been explored extensively in prior works on audio transformers (e.g., Septr[1], DTF-AT[2]). The contribution here appears to be mainly an adaptation of that concept to a spiking setting rather than a fundamentally new architectural design.
> **Response to Weakness-2.**
> We sincerely thank the reviewer for this thoughtful comment. We agree that the high-level idea behind SADA, namely a dual-branch time–frequency decoupling structure, has indeed been explored in prior ANN-based audio transformers such as SepTr [1] and DTF-AT [2]. In the revised manuscript, we will soften and adjust the description of our contributions so as not to over-emphasize the structural novelty of SADA itself.
>
> ---
>
> ### **1. Dual-branch structures are familiar in ANN audio, but this is the first application to SNN-based audio transformers**
>
> The time–frequency dual-branch structure itself is a well-established design pattern in ANN-based audio transformers [1, 2]. However, recent SNN foundation models and SNN-Transformers [3–5] have been almost exclusively developed for the vision domain. As a result:
> - They typically do not incorporate audio-specific time–frequency modules, and
> - They tend to reuse single-branch architectures originally designed for images directly on spectrogram inputs.
> In this sense, the goal of SADA is not to claim that we “invented” the dual-branch concept, but rather to:
> - Take a well-established audio design principle from ANNs (dual-branch time–frequency decoupling), and
> - Redesign and validate it under spike-based SNN constraints (binary spikes, event-driven updates, sparsity, spike accumulation, etc.).
>
> From this perspective, SADA plays a role analogous to early SNN-Transformer works that did not invent the Vision Transformer itself, but redesigned ViT-style architectures into spike-based transformers suited for SNNs. Those works are generally regarded as meaningful contributions in the SNN community, and we view SADA in a similar position for audio SNN-Transformers. Moreover, to the best of our knowledge, systematic time–frequency dual-branch architectures have not been explored in the context of SNN-based audio transformers: most existing SNN works focus on the vision domain, and our work can be viewed as one of the first attempts to extend Transformer-style SNN architectures to the audio domain with an explicit time–frequency decoupling design.
>
> ---
>
> ### **2. The core conceptual contribution is SDA; SADA is its audio-domain specialization**
>
> The central conceptual contribution of our paper is Spike Dice Attention (SDA). SDA explicitly analyzes and addresses the spike density-unaware problem that is commonly present in current spike-based self-attention formulations.
>
> As discussed in Section 3.3 and Appendix C, existing spike-based self-attention mechanisms (e.g., dot-product SSA, Hadamard-based SDSA) compute similarity purely based on spike counts (firing rates). Consequently:
>
> - A token can receive a very large attention score even if its true semantic similarity is not high,
> - Simply because its spike density is high.
>
> This behavior diverges from the intended behavior of ANN attention—namely, focusing on truly important tokens—and makes it difficult to faithfully reproduce ANN attention behavior when operating on spike tensors.
>
> To address this, SDA introduces a Dice-based spike similarity with spike-density normalization, which:
>
> - Suppresses tokens that are merely high-density, and
> - Assigns higher scores to “sparse but accurate matches” instead.
>
> We provide a mathematical analysis in Appendix C showing that SDA has a density-aware property, and that it yields attention scores that better approximate the functional behavior of ANN attention under SNN constraints.
>
> Empirically, as shown in Table 2 of the paper, simply replacing the spike-based attention in existing SNN-Transformers with SDA under the same training and evaluation settings yields consistent performance improvements on our audio benchmarks:
>
> - Spike-driven Transformer [3]: +0.012 mAP
> - Spike-driven Transformer V2 [4]: +0.027 mAP
> - QKFormer [5]: +0.002 mAP
>
> These results demonstrate that SDA quantitatively alleviates the density-unaware limitation of prior spike-based attention mechanisms.
>
> In this context, SADA is not positioned as a standalone, fundamentally new conceptual module, but rather as an audio-domain specialization of SDA:
>
> - The dual-branch time–frequency structure itself follows existing ANN audio transformers such as SepTr and DTF-AT [1, 2]
> - In each branch, we replace the attention with density-aware SDA
> - The entire module is designed to operate under spike-based, event-driven constraints as an audio SNN-Transformer block.

---

> ### Author Response · Authors · 2025-11-26
>
> ### **3. Why this adaptation is still meaningful in the SNN context**
>
> Even though dual-branch architectures are well known in ANN-based audio transformers, their behavior and effectiveness are not automatically guaranteed when ported to SNNs. For example, it is not obvious:
>
> - Whether the dual-branch structure remains effective when the input is binary spikes,
> - How spike-density–driven score inflation manifests in each branch,
> - How time and frequency representations behave under SNN-specific constraints (sparsity, temporal accumulation, reset dynamics, etc.), and
> - Whether SDA can operate stably within a dual-branch structure and still provide performance gains.
>
> Our experiments show that:
>
> - SADA (dual-branch SDA) achieves higher mAP than both single-branch SDA and the original SNN-Transformers on our audio benchmarks, and
> - Dual-branch time–frequency modeling is not only viable but can be advantageous in SNN-based audio transformers.
>
> Thus, we are not merely copying a known ANN design and applying it as-is. Instead, we:
>
> - Combine a newly proposed spike-based attention mechanism (SDA) with a dual-branch time–frequency structure, and
> - Perform, to the best of our knowledge, the first systematic design and validation of a dual-branch audio SNN architecture that respects SNN constraints.
>
> In the revised manuscript, we will:
>
> - Make it explicit that the dual-branch structure itself follows prior ANN works such as SepTr and DTF-AT [1, 2], and
> - Re-organize the contribution statement so that SDA is clearly emphasized as the main conceptual contribution, while SADA is positioned as an audio-specific realization that applies SDA to the SNN-Transformer setting.
>
> We hope that this clarification more accurately reflects the intended scope of our claims and addresses the reviewer’s concern regarding the novelty of SADA.
>
> ---
>
> **References (for this response)**
> [1] SepTr: Separable Transformer for Audio Spectrogram Processing. INTERSPEECH 2022.
> [2] DTF-AT: Decoupled Time-Frequency Audio Transformer for Event Classification. AAAI 2024.
> [3] Spike-driven Transformer. NeurIPS 2023.
> [4] Spike-driven Transformer V2. ICLR 2024.
> [5] QKFormer: Hierarchical Spiking Transformer using Q-K Attention. NeurIPS 2024.

---

> ### Author Response · Authors · 2025-11-26
>
> > **Weakness-3.**
> > It is unclear whether the Dice score operation used in SDA is hardware-friendly for neuromorphic deployment. Since hardware efficiency is a key motivation for SNNs, an analysis or discussion of its implementability on neuromorphic hardware would be valuable.
>
> **Response to Weakness-3.**
> We sincerely thank the reviewer for raising this important point. Since neuromorphic hardware efficiency is one of the core motivations for SNN research, we fully agree that it is crucial to analyze whether the Dice score operation used in SDA is compatible with spike-driven, hardware-friendly computation.
>
> To address this, we provide a dedicated, hardware-oriented discussion in our separate **Global rebuttal**, where we:
>
> - Show that both the numerator and denominator of the Dice score in SDA can be expressed as **spike-driven (AC-only) accumulations** over binary spike events, and therefore follow the same primitive operations (integer additions, logical ANDs, etc.) commonly assumed in neuromorphic SNN implementations; and
> - Explain how the scalar Dice normalization factor can be implemented using a **small lookup table (LUT)** or low-precision approximation, so that the overall pipeline can remain **integer/accumulate-only** without requiring full-precision division hardware.
>
> In the revised manuscript, we will also add a concise summary of this analysis in the main text, explicitly clarifying the neuromorphic implementability assumptions of SDA and pointing interested readers to the more detailed derivations and hardware-focused arguments. We kindly refer the reviewer to the Global rebuttal for the complete hardware-oriented discussion.

---

> ### Author Response · Authors · 2025-11-26
>
> > **Weakness-4.**
> > It is somewhat unexpected that DiceFormer surpasses pretrained ANN models such as AST on AudioSet, despite not using large-scale pretraining (e.g., ImageNet pretraining or SSL) and operating with only four timesteps. This raises concerns about training fairness and reproducibility. The authors are encouraged to clarify the experimental setup and indicate whether the code and pretrained checkpoints will be released to ensure transparency.
>
> **Response to Weakness-4.**
> We thank the reviewer for raising this important concern regarding fairness and reproducibility.
>
> ---
>
> ### **1. What DiceFormer is actually compared against (from-scratch vs. pretrained)**
>
> First, we would like to clarify a potential source of misunderstanding. As reported in Table&nbsp;1, we are **not** claiming that DiceFormer surpasses all pretrained ANN baselines on AudioSet. What we actually claim is:
>
> - When several ANN baselines are trained on AudioSet
>   **without** ImageNet pretraining or audio self-supervised (SSL) pretraining (i.e.,
>   trained **from scratch** on AudioSet labels only),
> - Then, under this **from-scratch** setting with no large-scale pretraining,
>   **DiceFormer achieves higher mAP than some of these ANN models.**
>
> The distinction between **from-scratch** and **pretrained** regimes is already encoded in the current table footnotes (e.g., “* ImageNet-pretrained”, “** audio self-supervised pretraining”, and `a / b(*,**) = from-scratch / pretrained`), but we agree that this notation can be confusing at first glance. In the revised manuscript, we will explicitly distinguish from-scratch vs. pretrained results in the main text and clearly state that we are **not** claiming that DiceFormer “beats” pretrained ANN models that benefit from additional large-scale pretraining.
>
> ---
>
> ### **2. Training setup and fairness of comparison**
>
> All DiceFormer models are trained on **AudioSet** using a **standard supervised setup with publicly available labels only**, and we do **not** use any external pretraining such as ImageNet initialization or SSL on additional audio corpora.
>
> From the perspective of comparison fairness, our intention is as follows. The ANN baselines are reported in two distinct regimes:
>
> - **Pretrained setting:** models that leverage ImageNet and/or audio SSL pretraining,
> - **From-scratch setting:** models trained from scratch on AudioSet labels only.
>
> Whenever we discuss that DiceFormer “performs better” than ANN baselines, this statement is **strictly restricted** to comparisons against **some from-scratch ANN baselines** trained under the same supervised label setting on AudioSet, and is meant only in the sense that DiceFormer attains higher mAP than those from-scratch ANN results. In contrast, the **pretrained** ANN results are presented as **strong upper baselines** that enjoy the advantage of large-scale pretraining that DiceFormer does not use, and we will explicitly acknowledge this in the revised text.
>
> ---
>
> ### **3. Reproducibility and release of code/checkpoints**
>
> Regarding reproducibility, all DiceFormer experiments on AudioSet are conducted under a **single, fixed training configuration**, including:
>
> - optimizer and learning-rate schedule,
> - batch size and number of training epochs,
> - data preprocessing and data augmentation, and
> - implementation details (e.g., model variants and training tricks).
>
> These settings are described in detail in **Appendix E.3** of the manuscript.
>
> Furthermore, if the paper is accepted, we will release the DiceFormer model code as well as the pretrained DiceFormer checkpoints trained on AudioSet.
>
> We hope that these clarifications help address the reviewer’s concerns about DiceFormer’s performance, fairness of comparison, and reproducibility.

---

### Author Response · Authors · 2025-11-19
**Global rebuttal: On the spike-driven and neuromorphic-friendly nature of SDA (1 / 4)**

We sincerely thank the reviewers for taking the time to evaluate our work.
The constructive questions and comments have helped us clarify and improve the manuscript.

> Weakness
> 1) It is unclear whether the Dice score operation used in SDA is hardware-friendly for neuromorphic deployment. Since hardware efficiency is a key motivation for SNNs, an analysis or discussion of its implementability on neuromorphic hardware would be valuable.
> 2) In your core formulation for Spike Dice Attention, Equation (4), both the numerator and the denominator consist of computed variables rather than raw spike events. This computational pattern does not adhere to the fundamental spike-driven paradigm inherent to Spiking Neural Networks (SNNs). Consequently, the mechanism as formulated is conceptually inconsistent with the principles of SNNs.
> 3) Prior SNN Transformers (e.g., Spikformer, Spike-driven Transformer, QKFormer) emphasize avoiding MACs in favor of AC-style primitives for spiking self-attention. In contrast, Eq. (4) introduces an input-dependent division (normalizing by the sum of activities), which is not offline-precomputable and is often unfriendly to neuromorphic substrates (division/reciprocal are costly, latency-prone, and may break integer/accumulate-only pipelines). So, my major concern is that such division-based operations fundamentally limit neuromorphic deployability. The paper should quantify and justify this choice.

The reviewers raised an important question regarding the implementability of the Dice normalization term on neuromorphic hardware.
Below, we clarify the computation model, show that SDA is fully spike-driven in both numerator and denominator, and explain why the normalization term does not violate neuromorphic constraints.
Previous SNN-Transformer works [1–3] define neuromorphic-friendly spike-driven computation as follows:
(1) All neural communication is represented by binary spike signals (0, 1);
(2) The computation is event-driven (no computation is performed when there is no input);
(3) \\(I_i[t] = \\sum_j (w_{i,j} \\cdot s_j[t])\\), where \\(s_j[t]\\) is a spike signal, \\(w_{i,j}\\) is a synaptic weight, and \\(I_i[t]\\) is the input current.

In summary, when the input current is expressed in a form where the weight is accumulated only when a spike is fired, such algorithms and architectures are defined as spike-driven.

Our Spike Dice Attention (SDA) is given by:
\\[
\\mathrm{SDS}(Q, K) = \\frac{2 \\cdot \\mathrm{sum}_d(Q \\odot K)}{\\mathrm{sum}_d(Q) + \\mathrm{sum}_d(K) + \\varepsilon} \\tag{4}
\\]
\\[
\\mathrm{Attn} = \\mathrm{SN}(\\mathrm{SDS}(Q, K)) \\odot V \\tag{6}
\\]

---

---

> ### Author Response · Authors · 2025-11-19
> **Global rebuttal: On the spike-driven and neuromorphic-friendly nature of SDA (2 / 4)**
>
> # Proof of spike-driven property of the numerator term
>
> We have $Q = \\mathrm{SN}(\\mathrm{BN}(X W\_{Q}))$, $K = \\mathrm{SN}(\\mathrm{BN}(X W\_{K}))$, and $V = \\mathrm{SN}(\\mathrm{BN}(X W\_{V}))$.
> From Figure 1, $X$ passes through a spiking neuron (SN) and becomes a spike signal consisting of 0s and 1s.
> Matrix multiplication with spike signals reduces to additions of the corresponding weights, which can be implemented as addressable additions on neuromorphic chips [1,2,4,5].
> The results then pass through SN again to produce spike signals $Q, K, V \\in \\{0,1\\}$.
>
> Therefore, in our numerator term $2 \\cdot \\mathrm{sum}\_{d}(Q \\odot K)$,
> the Hadamard product of the binary spike signals $Q$ and $K$ is simply an element-wise product of 0s and 1s,
> which is equivalent to an almost cost-free masking operation. Such masking can be implemented on neuromorphic chips
> using addressing algorithms or logical AND operations [1,2,5,6].
>
> In particular,
> $$
> 2 \\cdot \\mathrm{sum}\_{d}(Q \\odot K)
> = 2 \\cdot \\sum\_{d=1}^D Q\_{i,d}[t] \\odot K\_{j,d}[t],
> $$
> where $d$ is the channel index, $i, j$ are the query/key token indices, and $t$ is the time step.
> We can show the spike-driven property as follows:
>
> 1. $Q\_{i,d}[t] \\odot K\_{j,d}[t] \\in \\{0,1\\}$, which is effectively
>    $Q\_{i,d}[t] \\wedge K\_{j,d}[t]$ (logical AND).
>
> 2. For each channel $d$, define
> $$
> f\_{d}[t]
> = Q\_{i,d}[t] \\odot K\_{j,d}[t]
> = Q\_{i,d}[t] \\wedge K\_{j,d}[t].
> $$
>
> 3. From 1)–2), we have
> $$
> 2 \\cdot \\mathrm{sum}\_{d}(Q \\odot K)
> = 2 \\cdot \\sum\_{d=1}^D f\_{d}[t].
> $$
>
> 4. Let $w\_{i,j} = 1$.
>
> 5. From 1)–4), it follows that
> $$
> 2 \\cdot \\mathrm{sum}\_{d}(Q \\odot K)
> = \\sum\_{d=1}^D \\bigl(w\_{i,j} \\cdot f\_{d}[t]\\bigr).
> $$
>
> Therefore, the numerator term matches the spike-driven definition given in previous works [1–3].
>
> # Proof of spike-driven property of the denominator term
>
> From the above, we already have $Q, K \\in \\{0,1\\}$.
> Thus, $\\mathrm{sum}\_{d}(Q)$ and $\\mathrm{sum}\_{d}(K)$ follow the spike count + accumulate pattern used in existing SNN-Transformers [1,7],
> and can be implemented on neuromorphic hardware in the same way.
>
> For example,
> $$
> \\mathrm{sum}\_{d}(Q) = \\sum\_{d=1}^D Q\_{i,d}[t].
> $$
> If we define $w\_{i,d} = 1$, then
> $$
> \\mathrm{sum}\_{d}(Q) = \\sum\_{d=1}^D \\bigl(w\_{i,d} \\cdot Q\_{i,d}[t]\\bigr),
> $$
> which matches the spike-driven definition above.
> The term $\\mathrm{sum}\_{d}(K)$ can be derived in an identical spike-driven form.

---

> ### Author Response · Authors · 2025-11-19
> **Global rebuttal: On the spike-driven and neuromorphic-friendly nature of SDA (3 / 4)**
>
> # Proof of spike-driven property of the scaling term
>
> Our SDA is defined, for layer $l$, head $h$, query token $i$, and key token $j$, as:
> $$
> \\mathrm{SDS}(Q, K) = \\frac{2 \\cdot \\mathrm{sum}\_{d}(Q \\odot K)}{\\mathrm{sum}\_{d}(Q) + \\mathrm{sum}\_{d}(K) + \\varepsilon} \\tag{4}
> $$
> $$
> \\mathrm{Attn} = \\mathrm{SN}(\\mathrm{SDS}(Q, K)) \\odot V \\tag{6}
> $$
>
> More explicitly,
> $$
> \\mathrm{SDS}\_{i,j}^{(l,h)}[t]
> = \\frac{2 \\cdot \\sum\_{d} Q\_{i,d}^{(l,h)}[t] \\odot K\_{j,d}^{(l,h)}[t]}
>        {\\sum\_{d} Q\_{i,d}^{(l,h)}[t] + \\sum\_{d} K\_{j,d}^{(l,h)}[t] + \\varepsilon}.
> $$
>
> 2. Define the denominator as
> $$
> S\_{i,j}^{(l,h)}[t]
> = \\sum\_{d} Q\_{i,d}^{(l,h)}[t] + \\sum\_{d} K\_{j,d}^{(l,h)}[t] + \\varepsilon.
> $$
>
> 3. Then we can rewrite
> $$
> \\mathrm{SDS}\_{i,j}^{(l,h)}[t]
> = \\frac{2}{S\_{i,j}^{(l,h)}[t]} \\cdot
>   \\sum\_{d} Q\_{i,d}^{(l,h)}[t] \\odot K\_{j,d}^{(l,h)}[t].
> $$
>
> 4. Define the firing indicator
> $$
> f\_{d}[t] = Q\_{i,d}^{(l,h)}[t] \\odot K\_{j,d}^{(l,h)}[t].
> $$
> Since $Q\_{i,d}^{(l,h)}[t], K\_{j,d}^{(l,h)}[t] \\in \\{0,1\\}$, we have $f\_{d}[t] \\in \\{0,1\\}$.
>
> 5. Therefore,
> $$
> \\mathrm{SDS}\_{i,j}^{(l,h)}[t]
> = \\frac{2}{S\_{i,j}^{(l,h)}[t]} \\cdot \\sum\_{d=1}^D f\_{d}[t].
> $$
>
> Let the scalar coefficient be
> $$
> c\_{i,j}^{(l,h)}[t] = \\frac{2}{S\_{i,j}^{(l,h)}[t]}.
> $$
>
> 6. Then we obtain
> $$
> \\mathrm{SDS}\_{i,j}^{(l,h)}[t]
> = c\_{i,j}^{(l,h)}[t] \\cdot \\sum\_{d=1}^D f\_{d}[t].
> $$
>
> 7. In other words, the heavy computation along the channel dimension $D$ is still the
> AND + accumulate (popcount) operation over $f\_{d}[t]$, and Dice normalization only requires multiplying
> the scalar coefficient $c\_{i,j}^{(l,h)}[t]$ once for each $(i, j)$ pair and time step $t$.
> Thus, all channel-wise heavy operations are implemented as spike-driven computations
> (AND + spike count + accumulate), which is friendly to neuromorphic hardware.

---

> ### Author Response · Authors · 2025-11-19
> **Global rebuttal: On the spike-driven and neuromorphic-friendly nature of SDA (4 / 4)**
>
> # LUT-based implementation and memory overhead
>
> In addition, regarding the reviewer’s concern about the implementation cost of the division operation
> and the potential breakdown of an integer/accumulate-only pipeline,
> we emphasize that the scalar coefficient $c\_{i,j}^{(l,h)}[t]$ in the Dice normalization term
> can be implemented using a LUT (look-up table).
>
> Recall that
> $$
> S\_{i,j}^{(l,h)}[t] = \\sum\_{d} Q\_{i,d}^{(l,h)}[t] + \\sum\_{d} K\_{j,d}^{(l,h)}[t] + \\varepsilon
> $$
> is the spike count (popcount) at time $t$, so its value is restricted to a small integer range
> from 0 to $2D$.
> Regardless of the time index $t$, token indices $(i, j)$, or layer/head indices $(l, h)$,
> for any integer
> $$
> s \\in \\{1, 2, \\dots, 2D\\},
> $$
> we can precompute
> $$
> c(s) = \\frac{2}{s}
> $$
> and store it in a LUT using a 32-bit fixed-point format (4 bytes per entry).
>
> During inference, at each time step $t$, we take the spike count $S\_{i,j}^{(l,h)}[t]$ as the LUT index $s$,
> and obtain
> $$
> c\_{i,j}^{(l,h)}[t] = c\\bigl(S\_{i,j}^{(l,h)}[t]\\bigr)
> $$
> by a single LUT lookup. Therefore, no explicit division operation is required at runtime.
>
> Furthermore, Darwin3 [8] explicitly replaces division operations with LUT-based approximations when implementing SNN computations on neuromorphic hardware. More broadly, recent neuromorphic hardware systems for SNNs widely make use of LUT-based units for implementing diverse internal operations [8–11]. Therefore, implementing the Dice normalization scalar $ c(S) $ via a small LUT should be viewed as a natural and hardware-friendly design choice, rather than an unrealistic assumption that would break an integer/accumulate-only pipeline.
>
> ---
>
> # Spike-driven property of the DiceFormer architecture
>
> As shown in Figure 1 of the paper, except for the initial spike encoding stage,
> all neural communications in DiceFormer are carried by spike signals taking values in $\\{0,1\\}$.
> This means that DiceFormer operates in an event-driven fashion, just like previous SNN-Transformers [1–4,7]:
> when the input is 0, no computation is triggered.
> Consequently, matrix multiplications in the Conv and MLP blocks reduce to sparse additions,
> which can be implemented as addressable additions on neuromorphic chips [1,2,5].
>
> Once again, we sincerely thank the reviewers for their insightful comments,
> and we hope that our answers have addressed the concerns regarding the spike-driven and neuromorphic-friendly
> nature of our SDA and DiceFormer design.
>
> ## Reference
>
> [1] Spike-driven Transformer. In NeurIPS, 2023.
>
> [2] Spike-driven Transformer V2: Meta Spiking Neural Network Architecture Inspiring the Design of Next-generation Neuromorphic Chips. In ICLR, 2024.
>
> [3] SpikingResformer: Bridging ResNet and Vision Transformer in Spiking Neural Networks. In CVPR, 2024.
>
> [4] Spikformer: When spiking neural network meets transformer. In ICLR, 2023.
>
> [5] Event-driven Spiking Convolutional Neural Network. In WIPO Patent, 2020.
>
> [6] Towards Artificial General Intelligence with Hybrid Tianjic Chip Architecture. In Nature, 2019.
>
> [7] QKFormer: Hierarchical Spiking Transformer using Q-K Attention. In NeurIPS, 2024.
>
> [8] Darwin3: a large-scale neuromorphic chip with a novel ISA and on-chip learning. In National Science Review, 2024.
>
> [9] Efficient Synapse Memory Structure for Reconfigurable Digital Neuromorphic Hardware. In Frontiers in Neuroscience, 2018.
>
> [10] Beyond LIF neurons on neuromorphic hardware. In Frontiers in Neuroscience, 2022.
>
> [11] Large-Scale Neuromorphic Spiking Array Processors: A Quest to Mimic the Brain. In Frontiers in Neuroscience, 2018.

---

### Meta-Review · Area_Chair_zg2i · 2026-01-06

**Summary:**

This paper proposes DiceFormer, a spiking Transformer architecture for audio classification, introducing two main components: Spike Dice Attention (SDA), a density-aware spike-based attention mechanism inspired by Dice similarity, and Spike Audio Dice Attention (SADA), a time–frequency dual-branch extension tailored for audio spectrograms. The work is motivated by the observation that existing spike-based attentions suffer from a “spike-density–unaware” bias, leading to inflated attention scores for high-firing tokens.

Reviewers generally agreed that the paper is well executed, clearly motivated, and empirically strong, especially within the SNN audio domain, which is less explored than vision. The authors provided an exceptionally detailed rebuttal, including new experiments (CIFAR-100 vision validation, neuron-model ablations, ordering and ratio studies of SADA/SDA, hardware-oriented discussions, and fairness clarifications), and addressed almost every technical concern raised by the reviewers.

However, despite the high quality of the rebuttal and the substantial experimental effort, the discussion converged on several fundamental concerns that remain unresolved. These concerns relate to the conceptual novelty of the core idea, the degree of genuine theoretical advance beyond existing similarity measures and architectural adaptations, and lingering ambiguity around neuromorphic deployability as a central justification. These issues ultimately motivate the recommendation.

**Reviewer Concerns:**

Concerns largely addressed by the rebuttal
- Spike-driven and hardware-friendliness (etfE, T9Rj):
The authors provided a thorough derivation showing that SDA can be expressed via spike-driven accumulate-and-count operations, and argued that the Dice normalization can be implemented using LUT-based approximations consistent with prior neuromorphic systems. While not all reviewers were fully convinced, the rebuttal substantially clarified the intended hardware assumptions.
- Fairness of comparisons and neuron models (T9Rj, MSbv):
Additional clarifications and ablations (PLIF vs. LIF, LIF-only DiceFormer results) showed that the gains do not hinge on neuron choice, alleviating concerns about confounded comparisons.
- Ablations and architectural justification (4Zs6, T9Rj):
The authors added ratio and ordering ablations (e.g., 6:4, 4:6, reversed order), demonstrating that the chosen SADA→SDA (5:5) configuration is empirically justified rather than arbitrary.
- Generality beyond audio (MSbv):
CIFAR-100 experiments with drop-in SDA replacements showed modest but consistent gains on vision SNN Transformers, partially supporting generality claims.

Concerns that remain outstanding
- Limited conceptual novelty of the core mechanism:
While the authors frame “spike-density–unaware attention” as a newly identified issue, the proposed solution—Dice/F1-style normalization of overlap—draws directly from well-established similarity measures. The work is best characterized as a careful reformulation and engineering adaptation of known concepts to the spiking domain, rather than a fundamentally new attention principle. The additional analytical metrics (e.g., attention–density correlation) are useful diagnostics, but do not fully elevate the contribution to a new theoretical framework.
- Incremental nature of SADA relative to prior audio Transformers:
Even with clarifications that SADA is not claimed as a novel paradigm, the dual-branch time–frequency decoupling closely follows established ANN audio Transformers (e.g., DTF-AT, SepTr). Adapting this structure to SNNs is practically meaningful, but the resulting architectural contribution remains incremental, especially when presented as one of the main contributions.
- Neuromorphic motivation remains partially hypothetical:
Although the rebuttal convincingly argues that LUT-based normalization is possible, the paper does not provide concrete hardware synthesis, timing, or energy measurements demonstrating that SDA meaningfully improves real neuromorphic deployment compared to simpler spike-based attentions. As a result, the neuromorphic efficiency argument remains plausible but not decisive, and insufficient on its own to justify acceptance at a highly competitive venue.
- Contribution strength relative to ICLR standards:
Across reviews, a consistent pattern emerges: reviewers acknowledge strong engineering, careful experiments, and solid performance, but express hesitation that the work constitutes a sufficiently deep conceptual or theoretical advance beyond the current SNN-Transformer literature.

**Reviewer Scores:**

One reviewer strongly rejected the paper, expressing high confidence that the conceptual issues are fundamental. Other reviewers rated the paper as marginally below or slightly above threshold, stating that they “would not mind rejection.” While the rebuttal improved clarity and confidence in soundness, the overall enthusiasm for acceptance remains limited, particularly regarding novelty.

---

### Decision · Program_Chairs · 2026-01-26

Reject